# The effect of future ambient air pollution on human premature mortality to 2100 using output from the ACCMIP model ensemble

Raquel A. Silva[1], J. Jason West[1], Jean-François Lamarque[2], Drew T. Shindell[3], William J. Collins[4], Stig Dalsoren[5], Greg Faluvegi[6], Gerd Folberth[7], Larry W. Horowitz[8], Tatsuya Nagashima[9], Vaishali Naik[8], Steven T. Rumbold[7,*], Kengo Sudo[10], Toshihiko Takemura[11], Daniel Bergmann[12], Philip Cameron-Smith[12], Irene Cionni[13], Ruth M. Doherty[14], Veronika Eyring[15], Beatrice Josse[16], I. A. MacKenzie[14], David Plummer[17], Mattia Righi[15], David S. Stevenson[14], Sarah Strode[18,19], Sophie Szopa[20] and Guang Zeng[21,**]

[1]Environmental Sciences and Engineering, University of North Carolina, Chapel Hill, North Carolina, USA
[2]NCAR Earth System Laboratory, National Center for Atmospheric Research, Boulder, Colorado, USA
[3]Nicholas School of the Environment, Duke University, Durham, North Carolina, USA
[4]Department of Meteorology, University of Reading, Reading, United Kingdom
[5]CICERO, Center for International Climate and Environmental Research-Oslo, Oslo, Norway
[6]NASA Goddard Institute for Space Studies and Columbia Earth Institute, New York, New York, USA
[7]Met Office Hadley Centre, Exeter, United Kingdom
[8]NOAA Geophysical Fluid Dynamics Laboratory, Princeton, New Jersey, USA
[9]National Institute for Environmental Studies, Tsukuba, Japan
[10]Earth and Environmental Science, Graduate School of Environmental Studies, Nagoya University, Nagoya, Japan
[11]Research Institute for Applied Mechanics, Kyushu University, Fukuoka, Japan
[12]Lawrence Livermore National Laboratory, Livermore, California, USA
[13]Agenzia Nazionale per le Nuove Tecnologie, l'Energia e lo Sviluppo Economico Sostenibile (ENEA), Bologna, Italy
[14]School of GeoSciences, University of Edinburgh, Edinburgh, United Kingdom
[15]Deutsches Zentrum für Luft- und Raumfahrt (DLR), Institut für Physik der Atmosphäre, Oberpfaffenhofen, Germany
[16]GAME/CNRM, Meteo-France, CNRS—Centre National de Recherches Meteorologiques, Toulouse, France
[17]Canadian Centre for Climate Modeling and Analysis, Environment Canada, Victoria, British Columbia, Canada
[18]NASA Goddard Space Flight Center, Greenbelt, Maryland, USA
[19]Universities Space Research Association, Columbia, Maryland, USA
[20]Laboratoire des Sciences du Climat et de l'Environnement, LSCE-CEA-CNRS-UVSQ, Gif-sur-Yvette, France
[21]National Institute of Water and Atmospheric Research, Lauder, New Zealand
*Now at National Centre for Atmospheric Science (NCAS), University of Reading, Reading, United Kingdom
**Now at NIWA, Wellington, New Zealand

*Correspondence to*: J. J. West (jjwest@email.unc.edu)

**Abstract.** Ambient air pollution from ground-level ozone and fine particulate matter ($PM_{2.5}$) is associated with premature mortality. Future concentrations of these air pollutants will be driven by natural and anthropogenic emissions and by climate change. Using anthropogenic and biomass burning emissions projected in the four Representative Concentration Pathway scenarios (RCPs), the ACCMIP ensemble of chemistry-climate models simulated future concentrations of ozone and $PM_{2.5}$ at selected decades between 2000 and 2100. We use output from the ACCMIP ensemble, together with projections of future population and baseline mortality rates, to quantify the human premature mortality impacts of future ambient air pollution. Future air pollution-related premature mortality in 2030, 2050 and 2100 is estimated for each scenario and for each model using a health impact function based on

changes in concentrations of ozone and $PM_{2.5}$ relative to 2000 and projected future population and baseline mortality rates. Additionally, the global mortality burden of ozone and $PM_{2.5}$ in 2000 and each future period is estimated relative to 1850 concentrations, using present-day and future population and baseline mortality rates. The change in future ozone concentrations relative to 2000 is associated with excess global premature mortality in some

5 scenarios/periods, particularly in RCP8.5 in 2100 (316 thousand deaths/year), likely driven by the large increase in methane emissions and by the net effect of climate change projected in this scenario, but it leads to considerable avoided premature mortality for the three other RCPs. However, the global mortality burden of ozone markedly increases from 382,000 (121,000 to 728,000) deaths/year in 2000 to between 1.09 and 2.36 million deaths/year in 2100, across RCPs, mostly due to the effect of increases in population and baseline mortality rates. $PM_{2.5}$

concentrations decrease relative to 2000 in all scenarios, due to projected reductions in emissions, and are associated with avoided premature mortality, particularly in 2100: between -2.39 and -1.31 million deaths/year for the four RCPs. The global mortality burden of $PM_{2.5}$ is estimated to decrease from 1.70 (1.30 to 2.10) million deaths/year in 2000 to between 0.95 and 1.55 million deaths/year in 2100 for the four RCPs, due to the combined effect of decreases in $PM_{2.5}$ concentrations and changes in population and baseline mortality rates. Trends in future air

pollution-related mortality vary regionally across scenarios, reflecting assumptions for economic growth and air pollution control specific to each RCP and region. Mortality estimates differ among chemistry-climate models due to differences in simulated pollutant concentrations, which is the greatest contributor to overall mortality uncertainty for most cases assessed here, supporting the use of model ensembles to characterize uncertainty. Increases in exposed population and baseline mortality rates of respiratory diseases magnify the impact on premature mortality

of changes in future air pollutant concentrations and explain why the future global mortality burden of air pollution can exceed the current burden, even where air pollutant concentrations decrease.

## 1 Introduction

Ambient air pollution has adverse effects on human health, including premature mortality. Exposure to ground-level ozone is associated with respiratory mortality (e.g. Bell et al., 2005; Gryparis et al., 2004; Jerrett et al., 2009; Levy

et al., 2005). Exposure to fine particulate matter with aerodynamic diameter less than 2.5 µm ($PM_{2.5}$) is associated with mortality due to cardiopulmonary diseases and lung cancer (e.g. Brook et al., 2010; Burnett et al., 2014; Hamra et al., 2014; Krewski et al., 2009; Lepeule et al., 2012). Previous studies have estimated the present-day global burden of disease due to exposure to ambient ozone and/or $PM_{2.5}$ (e.g., Apte et al., 2015; Evans et al., 2013; Forouzanfar et al., 2015), with several studies estimating this burden using only output of global atmospheric models

(Anenberg et al., 2010; Fang et al., 2013a; Lelieveld et al., 2013; Rao et al., 2012; Silva et al., 2013). However, few studies have evaluated how the global burden might change in future scenarios (Lelieveld et al., 2015; Likhvar et al., 2015; West et al., 2007). Other global studies have estimated future air pollution-related mortality as a by-product of analyses of other future changes, such as the effects of climate change or of climate change mitigation (e.g., Fang et al., 2013b; Selin et al., 2009; West et al., 2013), but do not focus on the range of plausible future mortality as their

main purpose. Similarly, studies at local and regional scales have evaluated the mortality impact of changes in air quality due to future climate change (Bell et al., 2007; Chang et al., 2010; Fann et al., 2015; Heal et al., 2012;

Jackson et al., 2010; Knowlton et al., 2004, 2008; Orru et al., 2013; Post et al., 2012; Sheffield et al., 2011; Tagaris et al., 2009) but few such studies have evaluated changes beyond 2050.

Future ambient air quality will be influenced by changes in emissions of air pollutants and by climate change. Changes in anthropogenic emissions will likely dominate in the near-term (Kirtman et al., 2013 and references therein), and depend on several socio-economic factors including economic growth, energy demand, technological choices and developments, demographic trends and land use change, as well as air quality and climate policies. Climate change will affect the ventilation, dilution, and removal of air pollutants, the frequency of stagnation, photochemical reaction rates, stratosphere−troposphere exchange of ozone, and natural emissions (Fiore et al., 2012, 2015; Jacob and Winner, 2009; von Schneidemesser et al., 2015; Weaver et al., 2009). Climate change is likely to increase ozone in polluted regions during the warm season, particularly in urban areas and during pollution episodes. In remote regions, however, ozone is likely to decrease due to greater water vapor concentrations, which increase the loss of ozone by photolysis and subsequent formation of hydroxyl radicals (Doherty et al, 2013). The effects of climate change on $PM_{2.5}$ concentrations are generally uncertain as changes in temperature affect both reaction rates and gas to particle partitioning as well as wildfires and biogenic emissions, and vary regionally primarily due to differing projections of changes in precipitation (Fiore et al., 2012, 2015; Fuzzi et al., 2015; Jacob and Winner, 2009; von Schneidemesser et al., 2015; Weaver et al., 2009).

The Atmospheric Chemistry and Climate Model Intercomparison Project (ACCMIP) simulated preindustrial (1850), present-day (2000) and future (2030, 2050 and 2100) concentrations of ozone and $PM_{2.5}$ with an ensemble of 14 state-of-the-art chemistry climate models (Table S1) (Lamarque et al., 2013, Stevenson et al., 2013) to support the IPCC's Fifth Assessment Report. Using modeled 1850 and 2000 concentrations from this ensemble, we showed previously that exposure to present-day anthropogenic ambient air pollution is associated with 470 (95% Confidence Interval (CI): 140, 900) thousand deaths/year from ozone-related respiratory diseases, and 2.1 (1.3, 3.0) million deaths/year from $PM_{2.5}$-related cardiopulmonary diseases and lung cancer (Silva et al., 2013). These results were obtained for a wider range of cardiopulmonary diseases and using a different exposure-response model for $PM_{2.5}$ mortality than the present study, as discussed later.

The ACCMIP models simulated future air quality for specific periods through 2100, for four global greenhouse gas (GHG) and air pollutant emission scenarios projected in the Representative Concentration Pathways (RCPs) (Van Vuuren et al., 2011a and references therein). The four RCPs were developed by different research groups with different assumptions regarding the pathways of population growth, economic and technological development, and air quality and climate policies. Anthropogenic radiative forcing in 2100 ranges from a very low level in the mitigation scenario RCP2.6 (Van Vuuren et al., 2011b), to medium levels in the two stabilization scenarios, RCP4.5 (Thomson et al., 2011) and RCP 6.0 (Masui et al., 2011), to a high level in the very high baseline emissions scenario RCP8.5 (Riahi et al., 2011). All RCPs assume increasingly stringent air pollution controls as countries develop economically, leading to decreases in air pollutant emissions that reflect the different methods of the different RCP groups (e.g., Smith et al., 2011). But as assumptions are similar among the RCPs, the four scenarios do not span the range of possible futures published in the literature for short-term species. For example, other studies have simulated scenarios in which air pollution controls are kept at current levels while underlying trends (e.g., energy use) increase

overall emissions (Lelieveld et al., 2015; Likhvar et al., 2015). While most air pollutants are projected to decrease, ammonia increases in all RCPs due to the projected increase in population and food demand, and methane increases in RCP8.5 because of its projected rise in livestock and rice production. However, these scenarios follow different pathways in different regions. In some regions, emissions increase to mid-century before decreasing, while in other regions emissions are already decreasing at present and continue decreasing to 2100. Models in the ACCMIP ensemble incorporate chemistry-climate interactions, including mechanisms by which climate change affects ozone and $PM_{2.5}$, although models do not all include the same mechanisms of interactions and do not always agree on the net effect of these interactions (von Schneidemesser et al., 2015).

Using modeled ozone and $PM_{2.5}$ concentrations from the ACCMIP ensemble, we estimate the future premature human mortality associated with exposure to ambient air pollution. Our premature mortality estimates are obtained using a health impact function, combining the relative risk of exposure to changes in air pollution with future exposed population and cause-specific baseline mortality rates. We estimate overall future premature mortality considering the difference in air pollution associated with 2030, 2050 and 2100 emissions and climate relative to that resulting from 2000 emissions and climate. Mortality estimates are obtained at a sufficiently fine horizontal resolution (0.5°x0.5°) to capture both global and regional effects and inform regional and national air quality and climate change policy, but are not expected to capture local scale (e.g., urban) air pollution effects.

## 2 Methods

### 2.1 Ambient ozone and $PM_{2.5}$ concentrations

Concentrations of ozone and $PM_{2.5}$ in surface air are calculated for the present day (2000) and for the 2030, 2050 and 2100 decades for the four RCPs using the output of simulations by the ACCMIP ensemble of chemistry-climate models. As described by Lamarque et al. (2013) not all models are truly coupled chemistry climate models. OsloCTM2 and MOCAGE are chemical transport models driven by offline meteorological fields, and UM-CAM and STOC-HadAM3 do not model the feedback of chemistry on climate.

All ACCMIP models used nearly identical anthropogenic and biomass burning emissions for the present day and future, but they used different natural emissions (e.g. biogenic volatile organic compounds, ocean emissions, soil and lightning $NO_x$), which mostly impacted emissions of ozone precursors (Lamarque et al., 2013; Young et al., 2013) and natural aerosols (i.e., dust and sea salt). Model output shows good agreement with recent observations, both for ozone (Young et al., 2013) and for $PM_{2.5}$ (Shindell et al., 2013), although models tend to overestimate ozone in the Northern Hemisphere and underestimate it in the Southern Hemisphere, and to underestimate $PM_{2.5}$, particularly in East Asia. Future surface concentrations of air pollutants vary across scenarios and models, but ozone is projected to decrease except in RCP8.5, mostly associated with the large increase in methane concentrations specific to this scenario and the effect of climate change in remote regions (von Schneidemesser et al., 2015; Young et al., 2013).

We obtained hourly and monthly output from the ACCMIP ensemble simulations for a base year (2000) and for future projections under the four RCPs (2030, 2050 and 2100), with each time period corresponding to simulations

of up to 10 years, depending on the model. Only two models reported results for all four RCP scenarios and the three future time periods – GFDL-AM3 and GISS-E2-R. $PM_{2.5}$ is calculated as a sum of aerosol species reported by six models (see Supplemental Material), and four of these models also reported their own estimate of total $PM_{2.5}$ (Table S1). Our $PM_{2.5}$ formula includes nitrate; since this species was reported by three models only, we calculate the average nitrate concentrations in each cell reported by these models and add this average to $PM_{2.5}$ for the other models, following Silva et al. (2013). We use our $PM_{2.5}$ estimates to obtain all mortality results, and perform a sensitivity analysis using the $PM_{2.5}$ concentrations reported by four models using their own $PM_{2.5}$ formulas, which differed among models, as done by Silva et al. (2013). The native grid resolutions of the 14 models varied from 1.9°x1.2° to 5°x5°; we regrid ozone and $PM_{2.5}$ species surface concentrations from each model to a common 0.5°x0.5° horizontal grid to take maximum advantage of how the grids of different models overlap, following Anenberg et al. (2009, 2014) and Silva et al. (2013).

Ozone and $PM_{2.5}$ concentrations are calculated in each grid cell for each model separately. For both pollutants, we use identical metrics to those reported in the epidemiological studies we considered for the health impact assessment (next section):

- ▪ Seasonal average of daily 1-hr maximum ozone concentration, for the six consecutive months with highest concentrations in each grid cell;
- ▪ Annual average $PM_{2.5}$ concentration.

Among the 14 models, five models reported only monthly ozone concentrations, while the remaining models reported both hourly and monthly values. We calculate the ratio of the seasonal average of daily 1-hr maximum to the annual average of monthly concentrations, for each scenario/year, for those that reported both hourly and monthly concentrations. Then we apply that ratio to the annual average of monthly ozone concentrations for the former five models, as previously done by Silva et al. (2013). The differences in ozone and $PM_{2.5}$ concentrations between future year (2030, 2050 and 2100) and 2000 are shown in Tables S2 and S3, for each model. For ten world regions (Figure S1), we also estimate regional multi-model averages for each scenario/year (Figures S2 and S3).

**2.2 Health impact assessment**

We estimate future air pollution-related cause-specific premature mortality using generally the same methods as those used by Silva et al. (2013) to obtain present-day estimates, but with two important differences: (1) we use the recently published Integrated Exposure-Response (IER) model for $PM_{2.5}$ (Burnett et al., 2014) instead of a log-linear model (Krewski et al., 2009), and (2) we use projections of population and baseline mortality rates from the International Futures (IFs) integrated modeling system (Hughes et al., 2011).

We apply a health impact function to estimate premature mortality associated with exposure to ozone and $PM_{2.5}$ ambient air pollution *(ΔMort)* in each grid cell: *ΔMort = y0 \* AF \*Pop*, where *y0* is the baseline mortality rate (for the exposed population), *AF = 1 – 1/RR* is the attributable fraction, *RR* is the relative risk of death attributable to a change in pollutant concentrations, (RR=1 if there is no increased risk of death associated with a change in pollutant concentrations), and *Pop* is the exposed population (adults aged 25 and older). We calculate changes in premature mortality by applying the change in pollutant concentrations in each future year (2030, 2050 and 2100) relative to

year 2000 concentrations - the present-day state of air pollution - to the future population. To estimate ozone mortality, we apply the exposure-response function to the difference in ozone concentrations, while for $PM_{2.5}$ mortality we apply the exposure-response function to concentrations in each year (future years and 2000) and then subtract the mortality estimates. We therefore estimate 'avoided' / 'excess' premature mortality due to decreases / increases in air pollutant concentrations in the future years relative to 2000 concentrations. This approach differs from a calculation of the global burden of air pollution-related mortality since we use 2000 rather than 1850 concentrations as baseline. We estimate mortality changes due to future concentration changes, relative to the present, to avoid applying the health impact function at very low concentrations where there is less confidence in the exposure-response relationship. For example, the simulated 1850 air pollutant concentrations are often below the lowest measured value of the American Cancer Society study (Jerrett et al., 2009; Krewski et al., 2009). For illustration, we also estimate mortality relative to 1850 concentrations, which could be regarded as global burden of disease calculations, following Silva et al. (2013).

For each model, we estimate ozone-related mortality due to chronic respiratory diseases (RESP), using RR from Jerrett et al. (2009). We also estimate $PM_{2.5}$-related mortality due to ischemic heart disease (IHD), cerebrovascular disease (STROKE), chronic obstructive pulmonary disease (COPD) and lung cancer (LC), using RRs from the IER model (Burnett et al., 2014). We use RR per age group for IHD and STROKE and RR for all-ages for COPD and LC. We apply the IER model instead of RRs from Krewski et al. (2009), used by Silva et al. (2013), as the newer model should better represent the risk of exposure to $PM_{2.5}$, particularly at locations with high ambient concentrations. In the IER model, the concentration-response function flattens off at higher $PM_{2.5}$ concentrations yielding different estimates of excess mortality for identical changes in air pollutant concentrations in less-polluted vs. highly-polluted locations. Specifically, a one unit reduction of air pollution may have a stronger effect on avoided mortality per million people in regions where pollution levels are lower (e.g. Europe, North America, etc.) compared with highly-polluted areas (e.g. East Asia, India, etc.), which would not be the case for a log-linear function (Jerrett et al. 2009; Krewski et al. 2009). Therefore, using the IER model may result in smaller changes in avoided mortality in highly-polluted areas than using the log-linear model.

Each RCP includes its own projection of total population, but not population health characteristics. For all scenarios, we choose to use a common projection of population and baseline mortality rates per age group from the IFs (Figures S6 and S7). IFs projects population and mortality based on UN and WHO projections from 2010 through 2100, per age group and country, mostly based on three drivers – income, education, and technology (Hughes et al., 2011). Population projections from IFs differ from those underlying each RCP, but lie within the range of the RCPs (Figure S6). In 2030, global total population in IFs is within 0.08% of that reported for RCP2.6, RCP4.5 and RCP6.0 and 5% lower than for RCP8.5; however, in 2100 IFs projects larger global populations than RCP2.6 (+7%), RCP4.5 (+13%) and RCP6.0 (+2%) and considerably lower than RCP8.5 (-27%). IFs projects rising baseline mortality rates for cardiovascular diseases (CVD) and RESP, globally and in most regions (particularly in East Asia and India), reflecting an aging population. By using projections from IFs, we have a single source of population and baseline mortality rates, assuring their consistency and enabling us to isolate the effect of changes in air pollutant concentrations across the RCPs. Had we used the population projections from each scenario, the magnitude of the

changes (increases or decreases in premature mortality relative to 2000) would likely increase in RCP8.5, but decrease in RCP2.6, RCP4.5 and RCP6.0. With the exception of Europe, Former Soviet Union (FSU) and East Asia, where population is projected to decrease in 2100 relative to 2000, had we used present-day population and baseline mortality we would have obtained lower estimates for excess or avoided mortality in each scenario/year, as

projected increases in population and baseline mortality magnify the impact of changes in air pollutant concentrations. Therefore, we estimate the overall effect of future air pollution (due to changes in emissions and climate change) considering the population that will potentially be exposed to those effects. We also obtain different estimates of changes in future mortality than if we had calculated the global burden in each year, using air pollutant concentrations, population and baseline mortality rates in that year, and subtracted the present-day burden. Our

results do not reflect the potential synergistic effect of a warmer climate on air pollution-related mortality, i.e. we do not account for potential changes in the exposure-response relationships at higher temperatures (Pattenden et al. 2010; Wilson et al., 2014 and references therein).

Country-level population projections for 2030, 2050 and 2100 are gridded to 0.5°x0.5° using ArcGIS 10.2 geoprocessing tools, assuming that the spatial distribution of total population within each country is unchanged from

the 2011 LandScan Global Population Dataset at approximately 1 km resolution (Bright et al., 2012), and that the exposed population is distributed in the same way as the total population within each country. IFs projections of mortality rates for CVD are used to estimate baseline mortality rates for IHD and STROKE considering their present-day proportion in CVD (using GBD 2010 baseline mortality rates), as are RESP projections for COPD and malignant neoplasms for LC. IFs projections for 2010 are comparable to GBD 2010 (Lozano et al., 2012) estimates

for CVD (+0.04%), RESP (+2.5%) and neoplasms (-12%). We estimate the number of deaths per 5-year age group per country using the country level population. The resulting population and baseline mortality per age group at 30"x30" are regridded to the same 0.5°x0.5° grid as the concentrations of air pollutants.

Uncertainty from the RRs is propagated separately for each model-scenario-year to mortality estimates in each grid cell, through 1000 Monte Carlo (MC) simulations, i.e. we repeat the calculations in each grid cell 1000 times using

random sampling of the RR variable. For ozone, we use the reported 95% Confidence Intervals (CIs) for RR (Jerrett et al., 2009) and assume a normal distribution, while for $PM_{2.5}$ we use the values for the parameters alpha, gamma, delta and $z_{cf}$ (counterfactual) reported by Burnett et al. (2014) for 1000 MC simulations (GHDx 2013). Then for each of the 1000 simulations, we add mortality over many grid cells to obtain regional and global mortality and estimate the empirical mean and 95% CI of the regional and global mortality results. We assume no correlation

between the RRs for the four causes of death; thus we may underestimate the overall uncertainty for PM2.5 mortality estimates. Uncertainty in air pollutant concentrations is based on the spread of model results by calculating the average and 95% CI for the pooled results of the 1000 MC simulations for each model. This estimate of uncertainty in concentrations does not account for uncertainty in emissions inventories (as the ensemble used identical emissions) or for potential bias in modelled air pollutant concentrations. We also estimate the contribution

of uncertainties in RR and in air pollutant concentrations to the overall uncertainty in mortality estimates using a tornado analysis; we obtained global mortality estimates treating each variable as uncertain individually (year 2000 concentrations, future year concentrations, RR for ozone and the four parameters in the IER model for $PM_{2.5}$) and

used central estimates for all other variables, and then calculated the contribution of each variable to the overall uncertainty (when all variables are treated as uncertain simultaneously). Uncertainties associated with population and baseline mortality rates are not reported by IFs, and are not considered in the uncertainty analysis.

**3 Results**

First, we present our estimates of ozone and $PM_{2.5}$-related excess/avoided premature mortality in 2030, 2050 and 2100 for changes in pollutant concentrations between 2000 and each future period, for the four RCPs (sections 3.1 and 3.2, Figures 1 to 7). Figures 1 and 4 show global mortality for the different ACCMIP models. The multi-model average mortality results are shown for individual grid cells (Figures 2 and 5) and for regional totals (Figures 3 and 6). Finally, we include our estimates of the global mortality burden of both air pollutants for future concentrations relative to 1850 concentrations (section 3.3, Figures 8 and 9). In some cases, the changes in future mortality due to changes in future concentrations relative to 2000 show a different trend than the global mortality burden; this difference reflects the combined effects of future changes in concentrations relative to 1850, exposed population and baseline mortality rates.

**3.1 Ozone-related future premature mortality**

We find that future changes in ozone concentrations are associated with excess global premature mortality due to respiratory diseases in 2030, but avoided mortality by 2100 for all scenarios but RCP8.5 (Figure 1, Table S5). In 2030, all RCPs show excess multi-model average ozone mortality, ranging from 11,900 (RCP2.6)to 264,000 (RCP8.5) deaths/year. For each RCP, however, some models yield avoided mortality in 2030. In 2050, multi-model averages are obtained from only 3 or 4 models, depending on the scenario, which makes it difficult to compare with the other two periods. In 2100, we estimate excess ozone mortality in RCP8.5 (316,000 deaths/year), but avoided ozone mortality for the other three RCPs from -1.02 million (RCP2.6) to -718,000 (RCP6.0) deaths/year with all models agreeing in sign of the change.

Excess ozone-related future premature mortality (Figures 2 and 3, Table S6) is noticeable in some regions in 2030 for all RCPs, particularly in India and East Asia for RCP8.5 (over 95% of global excess mortality), but all scenarios except RCP8.5 show avoided global ozone-related mortality in 2100. Under this scenario in 2100, there are increases in ozone concentrations in all regions except North America, East Asia and Southeast Asia (Figure S2), likely driven by the projected large increase in methane emissions as well as by climate change. Avoided mortality in those three regions is outweighed by excess mortality in India, Africa and the Middle East. Also, some regions show different trends in future mortality relative to 2000 depending on the RCP, reflecting the effects of distinct assumptions in each RCP about economic growth and air pollution control with different trends in regional ozone precursor emissions. For example, North America and Europe show decreases in mortality through 2100 in all scenarios, except a slight increase in Europe for RCP8.5 in 2100. In East Asia, mortality peaks in 2050 for RCP6.0, driven by peak precursor emissions in 2050 in this scenario, but it peaks in 2030 for the other three RCPs. India shows peaks in mortality in 2050 followed by decreases for all RCPs but RCP8.5, in which mortality increases through 2100. Africa shows increases in mortality through 2100 for RCP2.6 and RCP8.5, while it peaks in 2050 for

RCP4.5 and decreases through 2100 for RCP6.0. Also, the effect of changes in population and baseline mortality rates is noticeable in some regions when comparing the trends in total ozone-related mortality and mortality per million people in each region (Figure S10). For example, decreases in population projected for 2100 in Europe, FSU and East Asia, are reflected in greater changes in mortality per million people than in total mortality, while the threefold increase in population in Africa amplifies the changes in total mortality.

For RCP8.5, we propagate input uncertainty to the mortality estimates (Figure 1, Table S45). Global future premature mortality changes from 264,000 (-39,300 to 648,000) deaths in 2030 to 316,000 (-187,000 to 1.38 million) deaths in 2100. Uncertainty in RR leads to coefficients of variation (CV) ranging from 31 to 37% (2030), 31 to 40% (2050) and 16 to 47% (2100) for the different models. Considering the spread of model results, overall CV for the multi-model average mortality increases to 66% (2030), 78% (2050) and 125% (2100). While uncertainty in RR and in modeled ozone concentrations have similar contributions to overall uncertainty in mortality results in 2050 (51% and 49%, respectively), in 2030 modeled ozone concentrations are the greatest contributor (81%), and in 2100 uncertainty in RR contributes the most to overall uncertainty (88%). For 2030, HadGEM2 differs in sign from the other 13 models with (avoided) global mortality totalling -33,900 deaths/year. For 2050, LMDzORINCA differs substantially from the other 3 models with -38,900 deaths/year. For 2100, HadGEM2 is a noticeable outlier with 1.2 million excess deaths/year and MOCAGE differs in sign from the other 12 models with -159,000 deaths/year.

### 3.2 PM$_{2.5}$-related future premature mortality

Global PM$_{2.5}$-related premature mortality, considering the difference in future concentrations and 2000 concentrations, decreases substantially in most scenarios, particularly in 2100 (Figure 4, Table S7). In 2030, the multi-model average varies from -289,000 (RCP4.5) to 17,200 (RCP8.5) deaths/year, although one model (CICERO-OsloCTM2) shows excess mortality for RCP2.6 and RCP8.5. In 2050, substantial avoided mortality is estimated for all scenarios except RCP6.0 which shows a small increase in mortality (16,700 deaths/year), but this is the average of only three models that do not agree on the sign of the change. In 2100, all scenarios show considerable avoided mortality, ranging from -1.31 million (RCP8.5) to -2.39 million (RCP4.5) deaths/year, reflecting the substantial decrease in emissions of primary PM$_{2.5}$ and precursors.

In several regions (North America, South America, Europe, FSU and Australia), PM$_{2.5}$ future premature mortality decreases through 2100 for all RCPs (Figures 5 and 6, Table S7). However, in East Asia, Southeast Asia, India, Africa, and the Middle East, for some scenarios, PM$_{2.5}$ mortality increases through 2030 or 2050 before decreasing. The changes in future mortality reflect changes in future PM$_{2.5}$ concentrations relative to 2000 (Figure S3), and a substantial increase in exposed population through the 21$^{st}$ century, particularly in Africa, India and the Middle East (Figure S6). That is, any reduction/increase in mortality due to the decrease/increase in pollutant concentrations was amplified by the increases in exposed population. The decreases in population in Europe, FSU and East Asia have similar effects as those mentioned above for ozone-related mortality. For example, while total avoided mortality in 2100 in East Asia decreases compared to 2050, for RCP2.6, RCP4.5 and RCP8.5, total avoided mortality per million people increases in the same scenarios (Figure S11). East and South Asia are the regions with the greatest projected

mortality burdens, and the variability in $PM_{2.5}$ among models is typically less in these regions than in several other regions globally, depending upon the scenario and year (Figure S9).

Future $PM_{2.5}$-related mortality estimates are influenced by the nonlinearity of the IER function. For example, in RCP8.5 in 2030, all models project an increase in global population-weighted concentration (Table S3) but all models except one show decreases in global $PM_{2.5}$-related mortality (Figure 4). This outcome results in part because $PM_{2.5}$ increases are projected in regions with high concentrations (particularly East Asia) that are on the flatter part of the IER curve, whereas $PM_{2.5}$ decreases in regions with low concentrations (North America and Europe) have a steeper slope and therefore a greater influence on global mortality.

Considering the results of the MC simulations for RCP8.5, premature mortality changes from -17,200 (-386,000 to 661,000) deaths in 2030 to -1.31 (-2.04 to -0.17) million deaths in 2100 (Figure 4, Table S7). Uncertainty in RR leads to a CV of 11 to 191% for the different models in the three future years. The spread of model results increases overall CV to 1644% (2030), 20% (2050) and 41% (2100). Uncertainty in modeled $PM_{2.5}$ concentrations in 2000 is the greatest contributor to overall uncertainty (59% in 2030, 45% in 2050, and 49% in 2100), followed by uncertainty in modeled $PM_{2.5}$ in future years (40% in 2030, 26% in 2050 and 32% in 2100). Uncertainty in RR has a negligible contribution to overall uncertainty in 2030 (<1%), as the multi-model mean mortality change happens to be near zero (one model projects a large increase while the other five models project decreases), but contributes 29% in 2050 and 20% in 2100.

We compared mortality results using our estimates of $PM_{2.5}$ from the sum of reported species with results using $PM_{2.5}$ reported by four models applying their own formula to estimate $PM_{2.5}$ (Figure 7). The multi-model average future avoided mortality for the four models which reported $PM_{2.5}$ is comparable although lower than the average for our $PM_{2.5}$ estimates for the same models. Individual models do not show the same differences in mortality using their own vs. our $PM_{2.5}$ estimates. Also, for two models (GFDL-AM3 and MIROC-CHEM) the two sources of $PM_{2.5}$ estimates yield mortality changes of different sign in 2030. These results reflect the different aerosol species included by each model to estimate $PM_{2.5}$ (e.g. nitrate is not included by all models).

**3.3 Global burden on mortality of ozone and $PM_{2.5}$**

Here we present estimates of the global burden on mortality of ozone and $PM_{2.5}$ concentrations in the future, considering the four RCPs relative to preindustrial concentrations (1850) and future exposed population and baseline mortality rates (Figures 8 and 9, Tables S8 and S9). For context, we estimate the present-day global burden, using 2000 concentrations, population from Landscan 2011 Population Dataset, and baseline mortality rates from GBD2010, to be: 382,000 (121,000 to 728,000) ozone deaths/year and 1.70 (1.30 to 2.10) million $PM_{2.5}$ deaths/year. These estimates are 18.7% lower for ozone-related mortality and 19.1% lower for $PM_{2.5}$-related mortality than those obtained in our previous study (Silva et al., 2013), reflecting; a) more restrictive mortality outcomes (chronic respiratory diseases rather than all respiratory diseases, and IHD+STROKE+COPD rather than all cardiopulmonary diseases); b) updated population and baseline mortality rates; c) the use of the recent IER model (Burnett et al., 2014) for $PM_{2.5}$ (instead of Krewski et al., 2009). Compared with the GBD 2013 (Forouzanfar et al. 2015), our estimates are 76% higher for ozone-related mortality and 42% lower for $PM_{2.5}$-related mortality, likely due to the

fact that we estimate the global mortality burden using 1850 concentrations as baseline, while Forouzanfar et al. (2015) consider counterfactual concentrations (theoretical minimum-risk exposure) that are mostly higher for ozone (uniform distribution between 33.3 and 41.9 ppb) and lower for $PM_{2.5}$ (uniform distribution between 5.9 and 8.7 $\mu g/m^3$) than 1850 concentrations. In addition, we consider ozone mortality from all chronic respiratory diseases while Forouzanfar et al. (2015) only account for COPD, and we restrict our mortality estimates to adult population while Forouzanfar et al. (2015) include $PM_{2.5}$ mortality from lower respiratory tract infections in children under 5 years old. As a sensitivity analysis, when we apply a counterfactual of 33.3ppb (instead of using 1850 concentrations), our ozone-related mortality estimates are 23% higher for the multi-model mean, varying between +10% and +52% among models. Similarly, using the IER model counterfactual, our $PM_{2.5}$-related mortality estimates are 22% lower for the multi-model mean, varying between -8% and -44% among models.

For ozone, the global mortality burden increases in all RCPs through 2050 to between 1.84 and 2.60 million deaths/year, and then it decreases slightly for RCP8.5 and substantially for the other RCPs, ranging between 1.09 and 2.36 million deaths/year in 2100. The increase can be explained by the rise in the baseline mortality rates for chronic respiratory diseases magnified by the increase in exposed population, while the decline is likely mostly related to the decrease in concentrations, slightly countered by further population growth (Figure 8). The global burden of mortality from $PM_{2.5}$ shows a declining trend for all RCPs from 2030 to 2100, peaking between 2.4 and 2.6 million deaths/year in 2030 then declining to between 0.56 and 1.55 million deaths/year in 2100, except for RCP6.0 which peaks in 2050 (3.50 million deaths/year) before declining considerably. For $PM_{2.5}$, the increase in exposed population and the decline in concentrations have a much greater effect than changes in baseline mortality rates (Figure 9). These results are similar to those of Apte et al. (2015) who report a stronger effect of projected demographic trends in India and China in 2030 than of changes in baseline mortality rates. Our estimates for the global burden of $PM_{2.5}$ mortality in 2050 (between 1.82 and 3.50 million deaths/year for the four RCPs) are considerably lower than those of Lelieveld et al. (2015) (5.87 million deaths / year for IHD+STROKE+COPD+LC), likely due to the assumption in the RCP scenarios of further regulations on air pollutants, while the Business-As-Usual scenario of Lelieveld et al. (2015) does not assume regulations beyond those currently defined.

To help explain differences between the trends in future global burden (Figures 8 and 9) and in future mortality relative to 2000 (Figures 1 and 4), we estimate the future global burden for two cases: Case A - using 2000 concentrations relative to 1850 and present-day population but future baseline mortality rates; and Case B – using 2000 concentrations relative to 1850 but future population and baseline mortality rates. Case A reflects the effect of future baseline mortality rates on the global burden, if concentrations in future years were maintained at 2000 levels, while Case B reflects the combined effect of population and baseline mortality rates, i.e. it is identical to Case A except that population changes. The difference between the global burden for each RCP and Case B reflects the effects of changes in future air pollutant concentrations, and nearly equals future mortality relative to 2000 concentrations in Figures 1 and 4. However, Cases A and B are calculated for all 14 models for ozone and 6 models for $PM_{2.5}$ (since all models reported air pollutant concentrations in 2000), while future mortality relative to 2000 is calculated for the models that report each scenario/year.

**4 Discussion**

In all RCP scenarios but RCP8.5, stringent air pollution controls lead to substantial decreases in ozone concentrations through the 21st century, relative to 2000. For RCP8.5, the higher baseline GHG (including methane) and air pollutant emissions lead to increases in future ozone concentrations. In contrast, global $PM_{2.5}$ concentrations show a decreasing trend across all RCP scenarios. These changes in air pollutant concentrations, combined with projected increases in baseline mortality rates for chronic respiratory diseases, drive ozone mortality to become more important relative to $PM_{2.5}$ mortality over the next century.

The importance of conducting health impact assessments with air pollutant concentrations from model ensembles, instead of from single models, is highlighted by the differences in sign of the change in mortality among models, and by the marked impact of the spread of model results on overall uncertainty in our mortality estimates. In most cases assessed here (ozone mortality in 2030 relative to 2000, $PM_{2.5}$ mortality in 2030, 2050 and 2100 relative to 2000), uncertainty in modeled air pollutant concentrations is the greatest contributor to uncertainty in mortality estimates. The differences in air pollutant concentrations reported by the ACCMIP models reflect different treatment of atmospheric dynamics and chemistry, chemistry-climate interactions, and natural emissions in each model (Young et al., 2013). Although there is likely a bias in estimating health effects using air pollutant concentrations from coarse resolution models (Li et al., 2015; Punger and West, 2013), particularly for $PM_{2.5}$, we do not expect resolution to be an important factor for the differences in simulated concentrations across coarse resolution models.

There are several uncertainties and assumptions that affect our results. We applied the same RR worldwide and into the future, despite differences in vulnerability of the exposed population, in composition of $PM_{2.5}$, and in other factors that may support the use of different risk estimates or different concentration-response relationships. These uncertainties can be addressed through additional long-term epidemiological studies, particularly for large cohorts in developing countries, to improve RR estimates globally. These studies should be representative of wider ranges of exposure and air pollutant mixtures than existing studies in the US and Europe, and they should control for confounding factors such as other environmental exposures, use of air conditioning, socio-economic factors, etc. Also, we estimate mortality for adults aged 25 and older, and do not quantify air pollutant effects on morbidity, so we underestimate the overall impact of changes in pollutant concentrations on human health. Uncertainty is evaluated for a single future population projection, not accounting for the wide range of projections in the literature, and does not reflect uncertainty in baseline mortality rates, as these are not reported; uncertainties in both population and baseline mortality rates would be expected to increase with time into the future. The spread of model results does not account for uncertainty in emissions inventories, as all ACCMIP models used the same projections of anthropogenic emissions. Moreover, climate and air quality interactions and feedbacks are not sufficiently understood to be fully reflected in modeled air pollutant concentrations, and global models simplify atmospheric physics and chemical processes. This is particularly important when modeling air quality given scenarios of future emissions and climate change. For example, most global models do not fully address climate sensitivity to biogenic emissions (e.g. isoprene, soil NOx and methane) and stratosphere-troposphere interactions (e.g. stratospheric influx of ozone). A better understanding of aerosol-cloud interactions, of the impact of climate change on wildfires, and of the impact of land use changes on regional climate and air pollution is also crucial.

Our results are limited by the range of air pollutant emissions projected by the RCPs, which assume that economic growth strengthens efforts to reduce air pollution emissions. All RCPs project reductions in anthropogenic precursor emissions associated with more extensive air quality legislation as incomes rise, except for methane in RCP8.5 and for ammonia in all scenarios. These scenarios together do not encompass the range of plausible air pollution futures for the 21$^{st}$ century, as the RCPs were not designed for this purpose (van Vuuren et al., 2011a). Other plausible scenarios have been considered, such as the Current Legislation Emissions and Maximum Feasible Reductions scenarios used by Likhvar et al. (2015) and the Business-As-Usual scenario of Lelieveld et al (2015). As noted above, our global burden estimates for 2050 are considerably lower than the Business-As-Usual scenario of Lelieveld et al. (2015). If economic growth does not lead to stricter air pollution control, emissions and health effects may rise considerably, particularly for scenarios of high population growth in developing countries (Amman et al., 2013).

**5 Conclusions**

Under the RCP scenarios, future PM$_{2.5}$ concentrations lead to decreased global premature mortality versus what would occur with fixed year-2000 concentrations, but ozone-related mortality increases in some scenarios/periods. In 2100, excess ozone-related premature mortality for RCP8.5 is estimated to be 316 thousand (-187 thousand to 1.38 million) deaths/year (likely due to an increase in methane emissions and to the net effect of climate change), while for the three other RCPs avoided ozone mortality is between -718 thousand and -1.02 million deaths/year. For PM$_{2.5}$, avoided future premature mortality is estimated to be between -1.33 and -2.39 million deaths/year in 2100. These reductions in ambient air pollution-related mortality reflect the decline in pollutant emissions projected in the RCPs, but the large range of results from the four RCPs highlights the importance of future air pollutant emissions for ambient air quality and global health. Mortality estimates differ among models and we find that, for most cases, the contribution to overall uncertainty from uncertainty associated with modeled air pollutant concentrations exceeds that from the RRs. Increases in exposed population and in baseline mortality rates of respiratory diseases magnify the impact on mortality of the changes in air pollutant concentrations.

Estimating future mortality relative to 2000 concentrations allows us to emphasize the effects of changes in air pollution in these results. However, increases in exposed population and in baseline mortality rates may drive an increase in the future burden of air pollution on mortality. Even in the most optimistic scenarios, the global mortality burden of ozone (relative to 1850 concentrations) is estimated to be over 1 million deaths/year in 2100, compared to less than 0.4 million in 2000. For PM$_{2.5}$, the global burdens in 2030 and 2050 for the four RCPs are greater than the global burden in 2000 but decrease to between 0.56 and 1.55 million deaths/year in 2100, compared to 1.7 million deaths/year in 2000. A strong decline in PM$_{2.5}$ concentrations for all RCPs together with demographic trends in the 21$^{st}$ century (with a projected substantial increase in exposed population) lead to a rising importance of ozone relative to PM$_{2.5}$ for the global burden of ambient air pollution-related mortality.

The RCPs are based on the premise that economic development drives better air pollution control, leading to improved air quality. This trend is apparent in some developing countries (Klimont et al., 2013), but it is yet to be determined how aggressive many developing nations will be in addressing air pollution. The assumed link

between economic development and air pollution control in the RCPs requires new and stronger regulations around the world, as well as new control technologies, for the air pollution decreases in the RCPs to be realized. The projected reductions in mortality estimated here will be compromised if more stringent policies are delayed (e.g., Lelieveld et al., 2015).

**Acknowledgements**

The research here described was funded by a fellowship from the Portuguese Foundation for Science and Technology, by a Dissertation Completion Fellowship from The Graduate School (UNC – Chapel Hill), and by NIEHS grant #1 R21 ES022600-01. We thank Karin Yeatts (Department of Epidemiology, UNC – Chapel Hill) for her help in researching projections of future population and baseline mortality rates, Colin Mathers (WHO) for

advising us on the IFs, Peter Speyer (IHME, University of Washington) for providing us access to GBD2010 cause-specific mortality data at the country-level, and Amanda Henley (Davis Library Research & Instructional Services, UNC – Chapel Hill) for facilitating our access to Landscan 2011 Global Population Dataset. The work of DB and PC was funded by the U.S. Dept. of Energy (BER), performed under the auspices of LLNL under Contract DE-AC52-07NA27344, and used the supercomputing resources of NERSC under contract No. DE-AC02-05CH11231.

Ruth Doherty, Ian MacKenzie and David Stevenson acknowledge ARCHER supercomputing resouces and funding under the UK Natural Environment Research Council grant: NE/I008063/1. GZ acknowledges the NZ eScience Infrastructure which is funded jointly by NeSI's collaborator institutions and through the MBIE's Research Infrastructure programme.

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

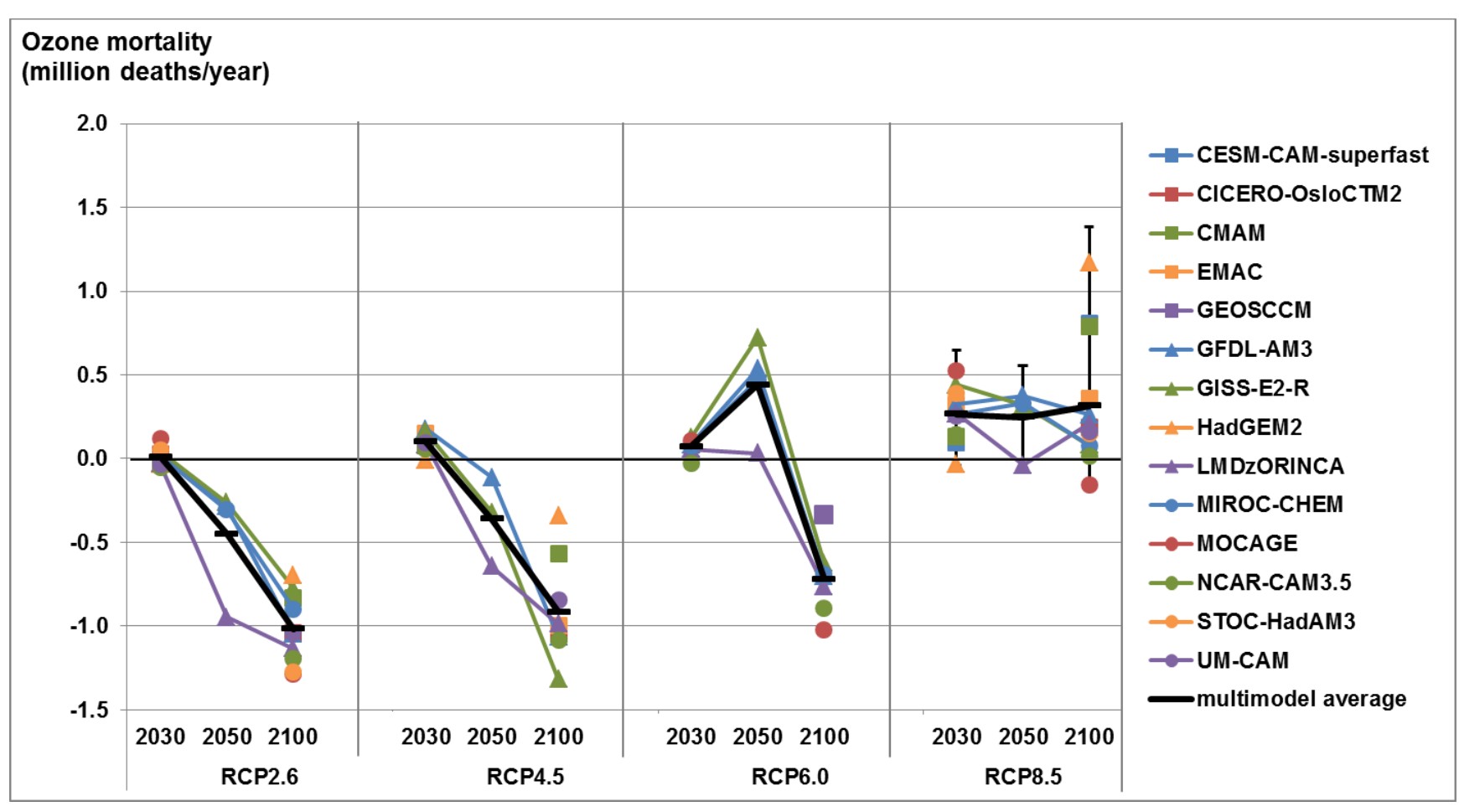

Figure 1: Estimates of future ozone respiratory mortality for all RCP scenarios in 2030, 2050 and 2100, showing global mortality for 13 models and the multi-model average (million deaths/year), for future air pollutant concentrations relative to 2000 concentrations. Uncertainty for the multi-model average shown for RCP8.5 is the 95% CI including uncertainty in RR and across models. Only models with results for the three years have lines connecting the markers.

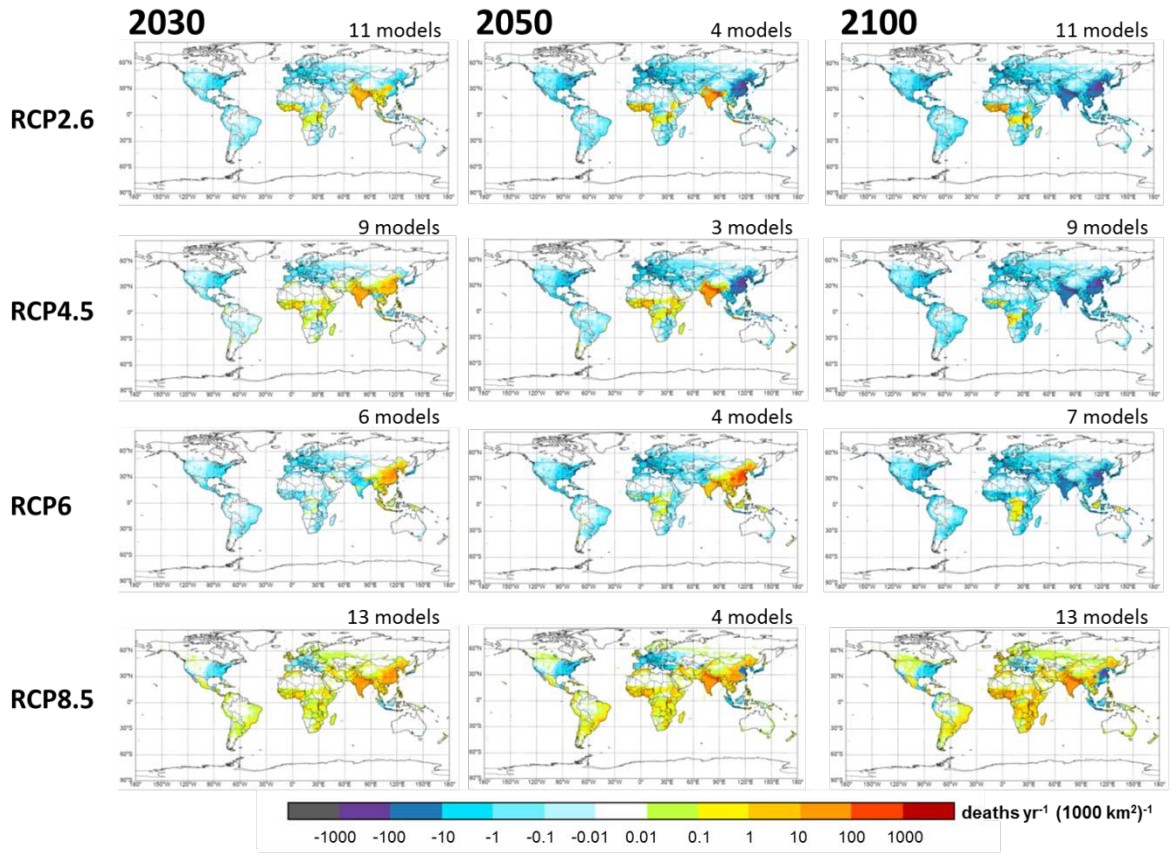

**Figure 2: Future ozone respiratory mortality for all RCP scenarios in 2030, 2050 and 2100, showing the multi-model average in each grid cell, for future air pollutant concentrations relative to 2000 concentrations.**

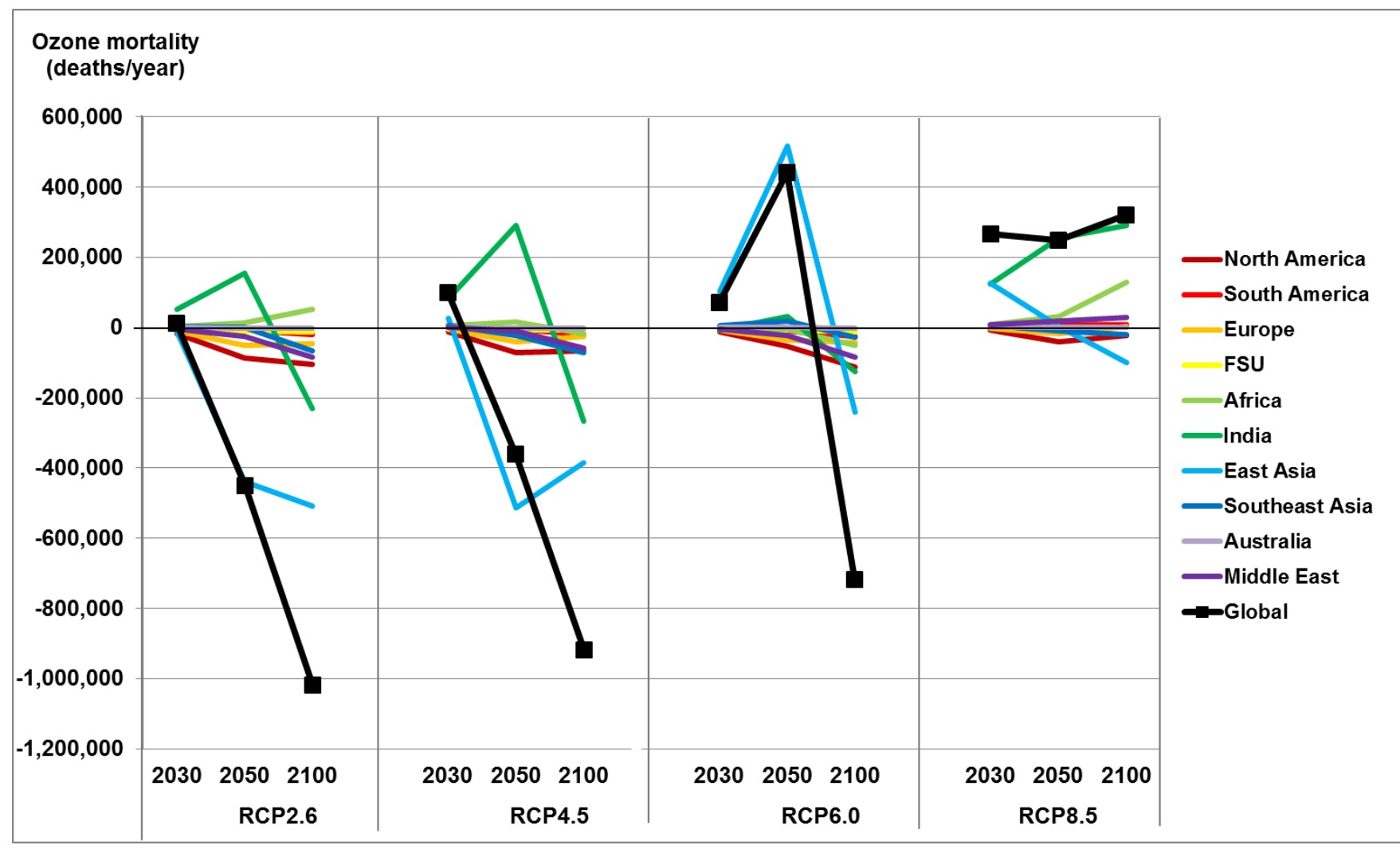

**Figure 3: Future ozone respiratory mortality for all RCP scenarios in 2030, 2050 and 2100, showing the multi-model regional average (deaths/year) in ten world regions (Figure S1) and globally, for future air pollutant concentrations relative to 2000 concentrations.**

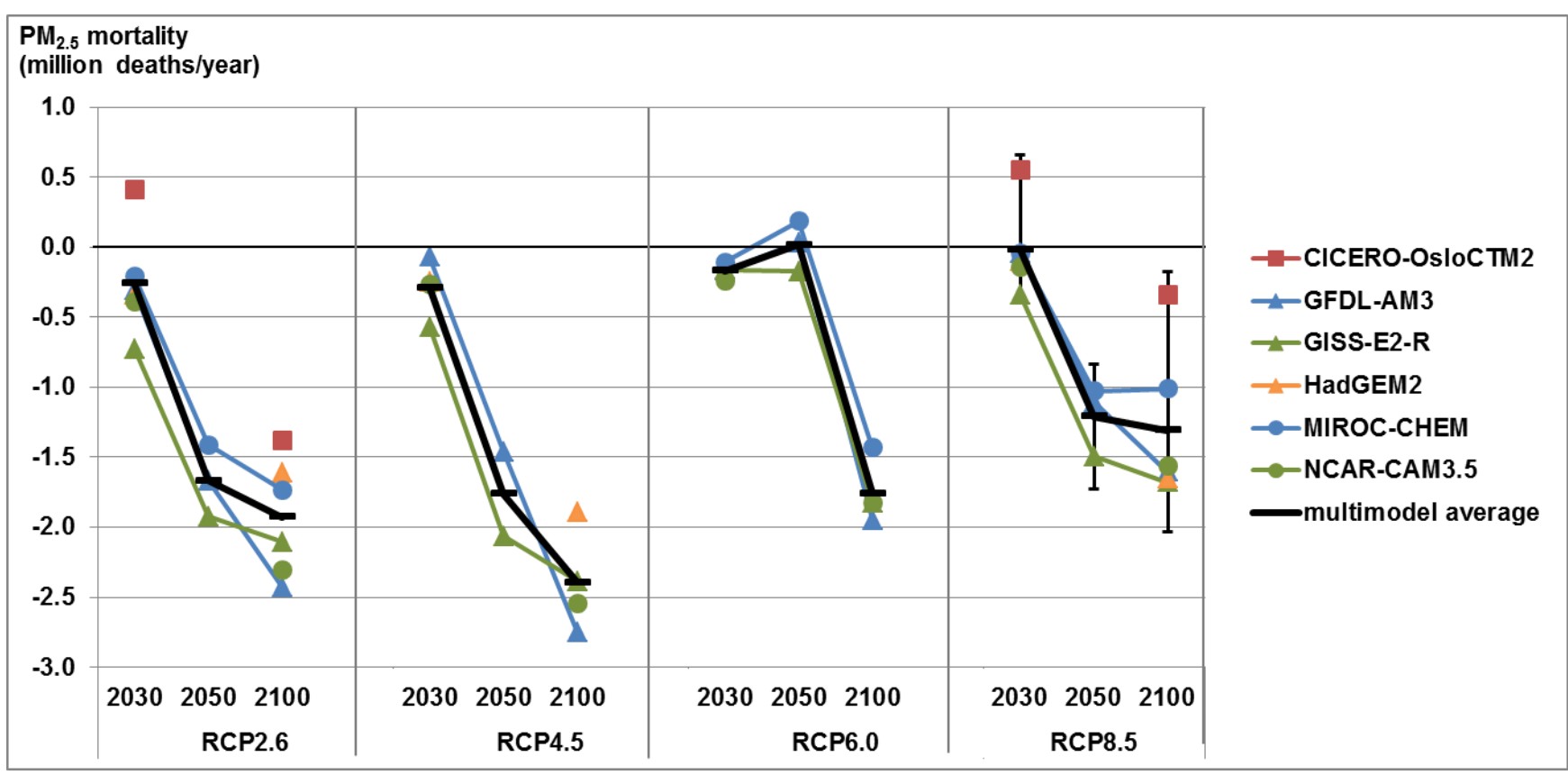

**Figure 4: Estimates of future premature mortality (IHD+STROKE+COPD+LC) for PM$_{2.5}$ calculated as a sum of species, for all RCP scenarios in 2030, 2050 and 2100, showing global mortality for six models and the multi-model average (million deaths/year), for future air pollutant concentrations relative to 2000 concentrations. Uncertainty shown for the RCP8.5 multi-model average is the 95% CI including uncertainty in RR and across models.**

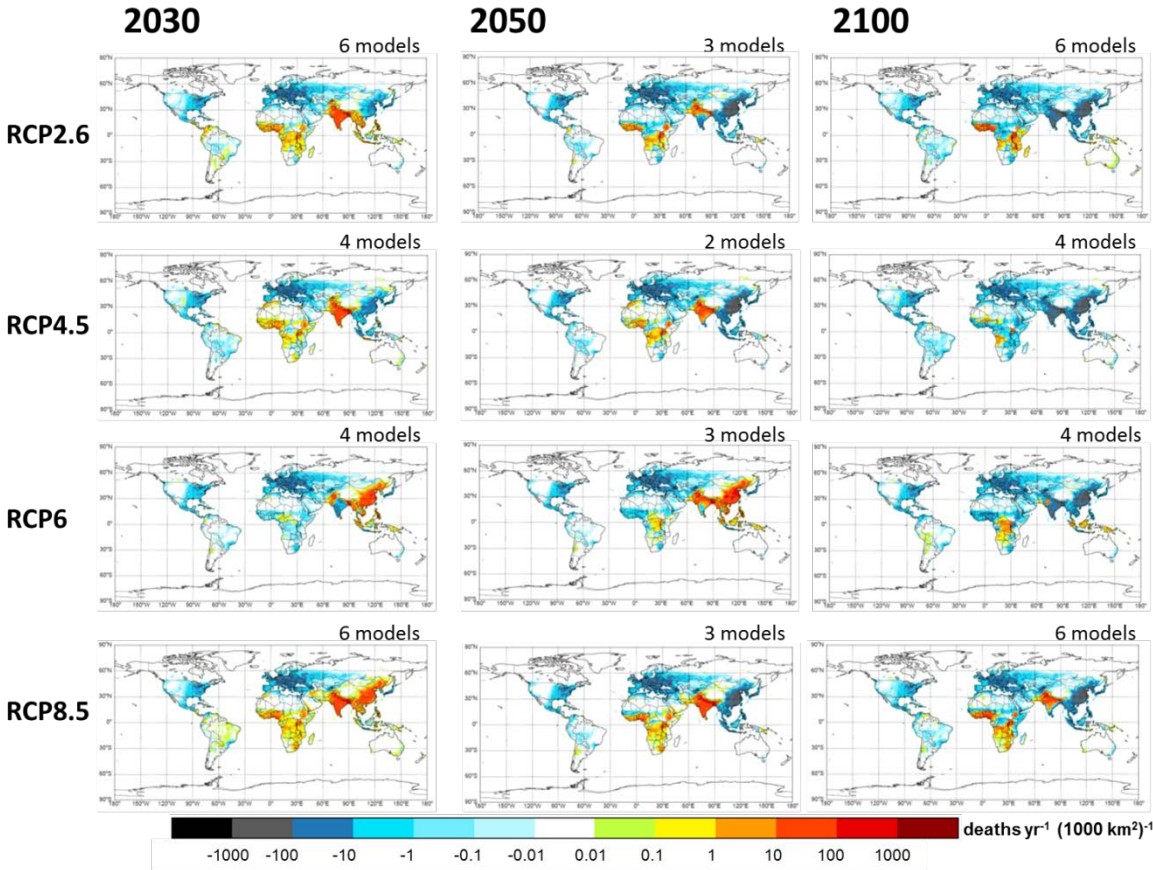

**Figure 5: Future premature mortality (IHD+STROKE+COPD+LC) for PM$_{2.5}$ calculated as a sum of species, for all RCP scenarios in 2030, 2050 and 2100, showing the multi-model average in each grid cell, for future air pollutant concentrations relative to 2000 concentrations.**

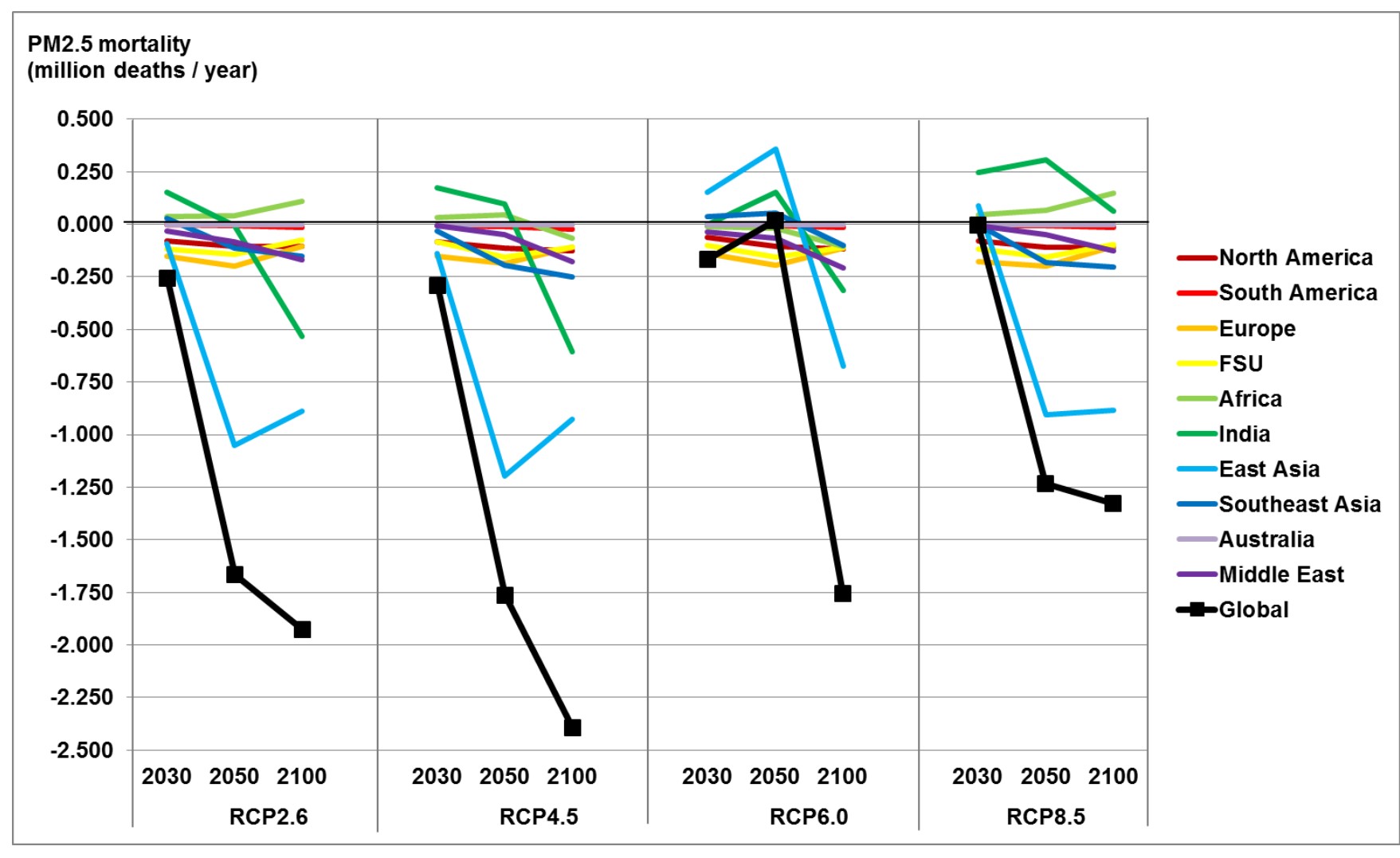

**Figure 6: Future premature mortality (IHD+STROKE+COPD+LC) for PM$_{2.5}$ calculated as a sum of species, for all RCP scenarios in 2030, 2050 and 2100, showing the multi-model regional average (deaths/year) in ten world regions (Figure S1) and globally, for future air pollutant concentrations relative to 2000 concentrations.**

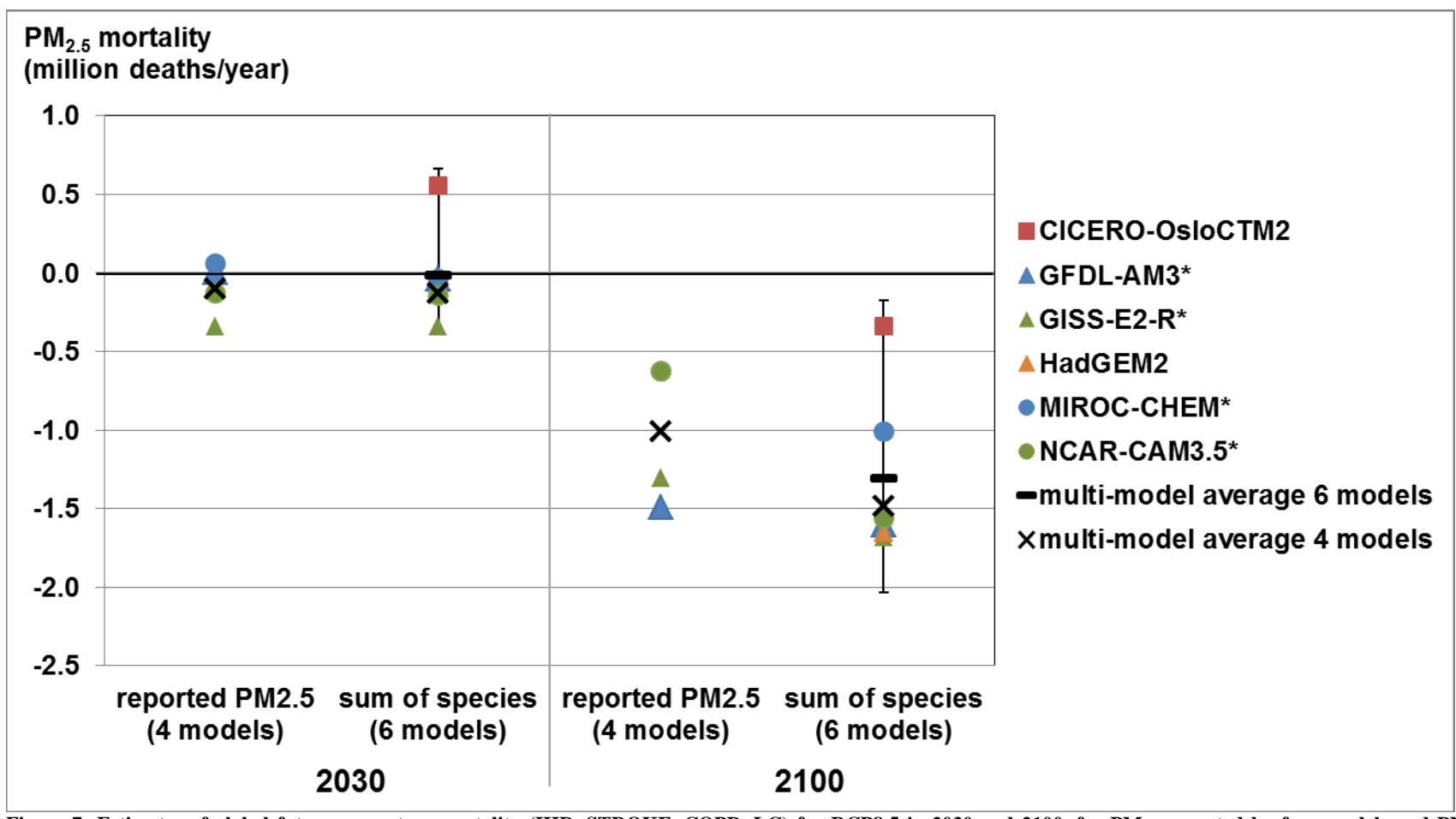

**Figure 7: Estimates of global future premature mortality (IHD+STROKE+COPD+LC) for RCP8.5 in 2030 and 2100, for PM$_{2.5}$ reported by four models and PM$_{2.5}$ estimated as a sum of species for six models, showing global mortality for each model and the multi-model average (million deaths/year), for future air pollutant concentrations relative to 2000 concentrations. Models signaled with * reported their own estimate of PM$_{2.5}$. Uncertainty shown for six models for sum of species is the 95% CI including uncertainty in RR and across models.**

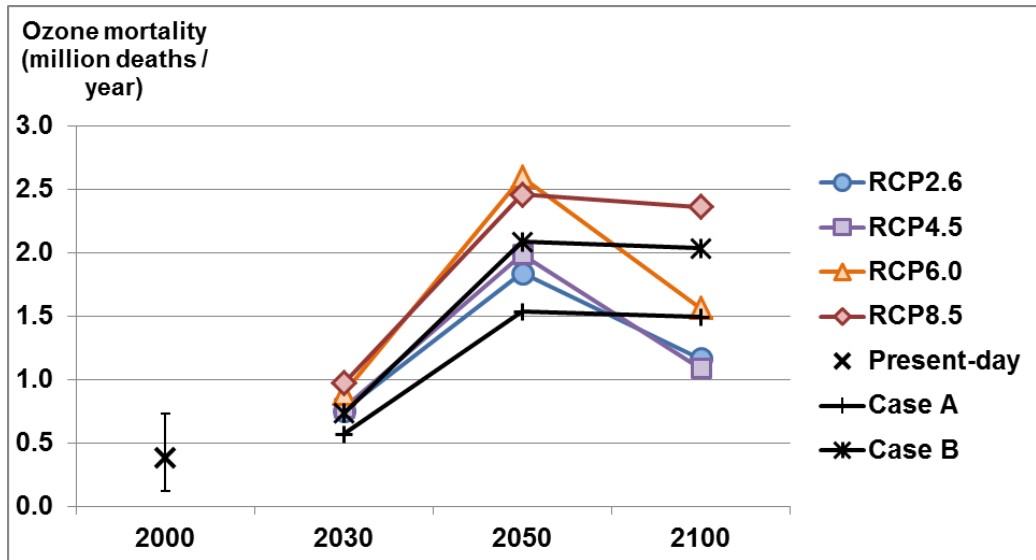

**Figure 8: Global burden on mortality of ozone concentrations relative to 1850, in the present day for 2000 concentrations, showing multi-model average and 95% CI including uncertainty in RR and across models (deaths/year), and in 2030, 2050 and 2100 for all RCPs, showing multi-model averages (deaths/year) given by the deterministic values. Also shown are future burdens using (Case A) 2000 concentrations relative to 1850 and present-day population but future baseline mortality rates and (Case B) 2000 concentrations relative to 1850 but future population and baseline mortality rates.**

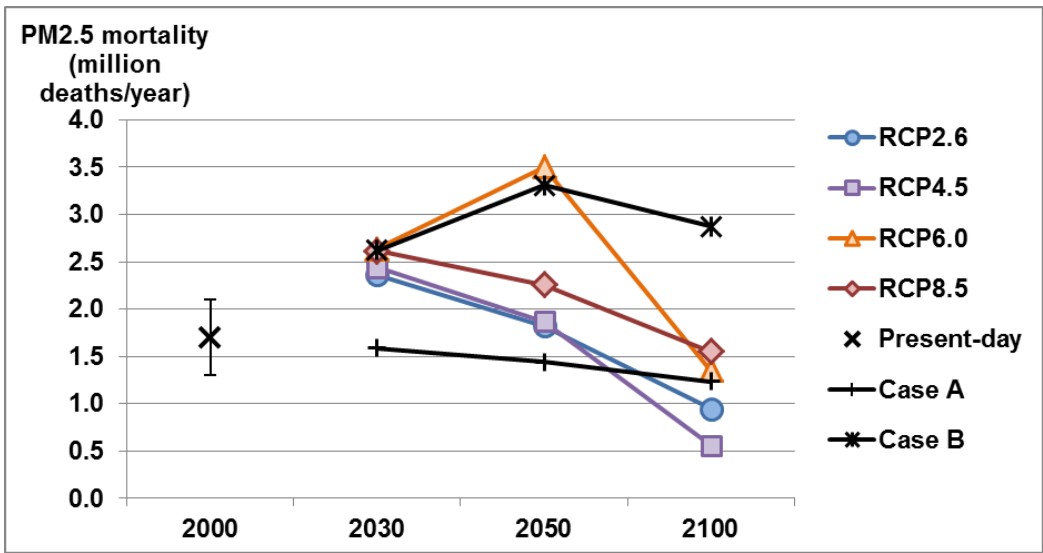

**Figure 9: Global burden on mortality of PM$_{2.5}$ concentrations relative to 1850, in the present day for 2000 concentrations, showing multi-model average and 95% CI including uncertainty in RR and across models (deaths/year), and in 2030, 2050 and 2100 for all RCPs, showing multi-model averages (deaths/year) given by the deterministic values. Also shown are future burdens using (Case A) 2000 concentrations relative to 1850 and present-day population but future baseline mortality rates and (Case B) 2000 concentrations relative to 1850 but future population and baseline mortality rates.**