# Peer review of "The effect of future ambient air pollution on human premature mortality to 2100 using output from the ACCMIP model ensemble"

_Atmospheric Chemistry and Physics, 2015_

## Referee Comment (RC1) · Anonymous Referee #3 · 3 Mar 2016

General comments: This study performs a global health impact assessment from ambient air pollution, using chemical transport or chemistry-climate models, for a set of RCP scenarios, for the years 2000, 2030, 2050 and 2100. Similar studies have been published before (properly acknowledged by the authors). The novelty of this study lies in the use of an ensemble of models, allowing for an evaluation of the contribution of model-calculated population exposure to pollution in the total uncertainty on the health impact. However a comparison of the outcome with previous studies, both for present day and future projections, is not obvious because of differences in methodology.

Specific comments: In the paper two ways are used to evaluate the impact of emission scenarios for the future on human health: 1) By using future demographics and

health statistics, and combining these with exposure to year 2000 pollutant levels and to pollutant levels corresponding to projected emissions for the specific year respectively and making the difference 2) by calculating the absolute number of mortalities for each considered year and making the difference with mortalities for 1850 ('mortality burden')

It took me a while to understand that reported 'avoided' and 'excess' mortalities refer to method 1). It should be better explained in the methodology section. Usually, avoided or excess mortalities for a given scenario are calculated versus a reference scenario for the same year (e.g. a stringent policy versus a business-as-usual as reference case). It's not clear here what the year 2000 pollution transposed to 2030 and 2050 actually represents as a reference. The avoided or excess mortalities can not be directly linked to specific policies (which pathway would have led to the year 2000 levels in 2030 - 2050 - 2100?). Wouldn't it make more sense to use e.g. RCP 8.5 as a reference, and evaluate the benefits of the 2.6 and 4.5 pathways? Using year 2000 pollution levels as a reference for future years also introduces an issue with exposure; concentration field spatial distribution is linked to population spatial patterns – in particular for PM. Does is make sense to overlay year 2000 pollution spatial patterns with year xxxx population spatial distribution?

Mortalities are estimated at 0.5x0.5 deg resolution: is this just a regridding of the native model resolution or was any downscaling done to better estimate the exposure in densely populated areas? Apparently the concentrations are just regridded; this can not be considered as a proper population-weighted exposure estimate at the coarse resolution of the models, as all population within a single grid will be exposed to the same level.

Regarding the use of Burnett's IER functions: specify whether age-specific functions have been used or all-ages. From what is written in the first par. of page 15, I understood that the Burnett functions have been applied without the counterfactual value? In fact it is not well explained how teh difference with 1850 was made: by first subtracting

1850 concentrations and then applying the exposure-response functions, or by applying exposure-response functions to both years and then subtracting mortalities. And how was it done for calculating the excess/avoided mortalities relative to year 2000?

The numbers in Table S3 do not seem to be consistent with year 2030 mortalities in Figure 4: In Table S3 only 2 models predict a global mean decrease in PM2.5 for RCP2.6 in 2030. In Figure 4 all models except 1 show a decrease in mortalities by 2030...Similar for the other RCPs; most flagrant for RCP8.5 where all PM2.5 appears to increase globally but only 1 model leads to an increase in mortality. How to explain this?

Table S4: should be mentioned as 'CHANGE' in mortalities between year 2000 pollution levels and respective scenario/year pollution levels. Also on Page 11, "Global 309 future premature mortality rises from 264,000 (-39,300 to 648,000) deaths in 2030 to 316,000 (-310 187,000 to 1.38 million) deaths in 2100" may cause confusion as these are again changes compared to 2000 pollution levels.

The fact that the range spans from negative to positive implies that the result is not significantly different from 0?

What has been the benefit of the multi-model analysis? And what can be learned from analyzing the RCP scenarios? Are the outcomes plausible in the light of the implicitly assumed rather stringent pollution controls?

The results section is dry and hard to digest with long lists of numbers of mortality changes per scenario, per region, with differences between models – all things that are much easier to read from the figures than in the text. For the reader it is hard to keep an overview and grasp the major message. Suggest to reduce and condense this section to most salient observations that are maybe not directly evident from the figures.

Discussion section: it looks like there is an increasing relative importance of O3 as health impact compared to PM for the future (what is the relative contribution of each

pollutant to total pollution mortality burden in each year, each scenario?) – this may be worth a few lines of discussion.

It is surprising that for the same emission scenarios, models have such different outcomes. Does the resolution play a role here? What could be done to improve the exposure estimate? Downscaling techniques? Use of regional models? Is it possible to evaluate the error made by using course resolution models?

It would be nice to see a graph summarizing other paper's results and this one (with error bars) for projected mortality burdens and to discuss what could be learned from this comparison.

---

## Referee Comment (RC2) · Anonymous Referee #1 · 11 Mar 2016

This is a valuable analysis on an important topic that takes advantage of a detailed model ensemble to address air pollution mortalities. However, the work has a few major deficiencies which ought to be addressed in a revision, as well as several minor comments.

1) The paper falls short in putting the work in the context of previous efforts. It cites only two previous papers assessing global health-related impacts of future air pollutants, while there is in fact a larger literature, including both global and regional effects. By not putting their work in the context of the previous work, the authors overstate the novelty of their contribution, and are not able to discuss their work in the context of what is known or unknown in this field.

[Figure]

2) The paper does not fully take advantage of the potential utility of using a multi-model ensemble in the analysis. As is, it presents straightforward calculations of resulting mortalities, which (for the most part) are very predictable based on the air quality results presented in previous work. The authors do have some very interesting results about the relative contributions of different assumptions of air pollution concentrations, exposure-response functions, population, etc. While some of these results are noted, in my opinion, they are the most interesting implications of this analysis and could be highlighted. However, the authors fall short in this area by not correctly characterizing the uncertainties and variabilities captured by their use of the RCP scenarios (a significant limitation which could be more thoroughly discussed and caveated) and the ACCMIP effort. These ought to be discussed more carefully.

3) The mortality numbers, while interesting, are not put in proper context such that the reader can understand what they mean. To address this, some comparison with existing literature could be very useful.

In addition to these major comments, there are several areas in which analyses are not fully described, and/or relevant methods-related information is missing. These are noted below.

Minor comments follow:

p3, line 30-31: "few studies have evaluated how the global burden might change in future scenarios" — this seems like a small slice of the literature. There are other papers that could be cited here.

p3, line 26-27: "RCPs. . .do not span the range of possible futures published in the literature for short-term species." This is a key point and it could be highlighted.

p4, line 4-6: but the ACCMIP is coarser. The mortality estimates thus should be justified. Also, line 33-34 on same page: this regridding to a scale finer than that modeled should be better described and justified.

p5, line 19-20: "similar to Silva et al 201...except for..." Does this mean exactly the same as the Silva et al 2013 paper except for those two differences? The description is unclear, and the language here could be more precise.

p6, line 9-10: using a common projection of population across the RCPs introduces both consistency in this analysis, but inconsistency relative to underlying social drivers. The implications of this choice should be discussed further, with quantifications of the magnitude as well as the direction.

p6, line 27-28: I can guess what the authors are referring to here, but the language could be easily misinterpreted (as the authors do actually look at the influence of climate on air pollutants themselves, just not modifications in ER factors). Rephrase?

p7, line 1-5: are potential correlations between different RRs accounted for in the Monte Carlo sampling? If so, how is that done? If not, the spread could be artificially narrowed. Please discuss.

p7, line 10: for the ACP audience, please describe 'tornado analysis' more thoroughly and quantitatively. Also, it is not addressed again, and there is no associated figure that corresponds to a traditional tornado-type plot.

p7. line 7: While the authors do have a certain spread of air pollutant concentrations, this should not be taken as a measure of 'uncertainty'. It is decidedly not a quantitative uncertainty analysis, as there are many other factors affecting 'uncertainty' in air pollutant concentrations that are not captured by the ACCMIP ensemble. This should be noted and discussed, and language carefully examined throughout the paper.

p 7, line 19: "In some cases..." This sentence is confusing. Rephrase?

p 9, lines 21-22: I'm not clear what was done here. This should be addressed in detail in methods.

p 10, line 31-32: This difference is noted. However, anyone familiar with the ACCMIP effort could have gleaned this simply from the previous reported results. What is new

here? Why is this particularly significant in terms of mortality?

p 11, line 16+ This could be discussed in more depth, including more quantitatively, as it's a key limitation of the authors' analysis.
* * *

---

## Referee Comment (RC3) · Anonymous Referee #2 · 18 Mar 2016

This manuscript uses the RCPs to project estimated air pollutant levels and health impacts globally for 10-year intervals between 2000 and 2100. It advances previous publications through the use of projected baseline mortality and population size along with projected air pollutant concentrations and therefore one can isolate the impacts of projected emissions from those of demographic changes in estimating future health impacts from air pollution. Further, the use of ensemble forecasts allows for the evaluation of the role of model variability in future estimates. Interestingly, while mortality impacts related to PM2.5 levels are projected to decrease under all scenarios, mortality from ozone exposure is projected to increase in all scenarios due to changes in population demographics, the absence of widespread decreases in ozone concentrations, increases in methane and climate warming. There are two main analyses in the manuscript:1) the impact of concentration changes relative to those in the year 2000 which is focused on the effects of future emissions and the variability between the different models, and 2)the assessment of the overall burden of disease attributable to air pollution in future years relative to pre-industrial (1850) concentrations where the relative impacts of emissions, and population projections are compared (cases A and B).

General comments Overall the manuscript provides unique new information to assess both potential future health impacts under well-defined scenarios and the role of model variability, uncertainty in concentration-response functions, uncertainty in emissions and the role of demographic changes in the estimation of future impacts. While the absolute numbers from the simulations are interesting, arguably more important is the assessment of uncertainty and the relative roles of different factors (demographics, emissions) in future estimates. For this component of the manuscript, decreasing the emphasis on the absolute numbers while providing more relative comparisons would help the reader sort through all of the results. Further the manuscript would benefit from some clear take-home messages on the relative impacts of future emissions and demographic changes and on the largest contributors to overall uncertainty. This information is in the manuscript but is hard to find and needs to be brought forward (even if it means decreasing emphasis on the absolute numbers).

The estimates for 2000 are low compared with other similar estimates and the authors attribute this to the choice of counterfactual. Given that the counterfactual is a choice, it would seem useful to isolate the impact of the choice of counterfactuals if the absolute number is being emphasized – some simple sensitivity analyses in which, for example, the Global Burden of Disease counterfactuals were applied, would be useful.

Future ozone and PM2.5 attributable mortality is clearly driven by China and India; given this it might be useful to present (or at least comment on) the model variability in these regions as what appears to be overall agreement across most of the models

may be a result of smoothing due to other regions which have relatively minor impacts on future trends.

Specific comments Abstract should be more consistent in presenting uncertainty in estimates and should include some quantification of uncertainty. Abstract should also provide more emphasis on uncertainty and relative impacts of different sources for the burden of disease estimates

L89 -Lim et al should be updated with Forouzanfar et al., 2015

L102 - suggest that in future ozone concentrations will decrease with climate change; can this be reconciled with observations on global increases during recent periods? (Emissions vs warming?)

L239 How do IF projections compare with current numbers, i.e. from the Global Burden of Disease ($\sim$for 2010)?

L283 –Should mention in limitation/discussion that the absence of uncertainty in the IF projections may be as important as other sources of uncertainty and that this uncertainty would increase over time (i.e. 2100 vs 2030)

L299-310 –There would appear to be $\sim$20x variability estimates for the different RCP scenarios - this is very large and clearly makes the case that emissions DO matter - it seems that this point should also be brought out a bit more.

L404 what are the 1850 concentrations that are used as the counterfactual? These should eb provided in the text.

Apte JS et al., ES&T 2015 also estimates future mortality assuming only changes on population – it would be useful to cite this paper and make some rough comparisons

L472 "preature" typo

---

## Author Comment (AC1) · 26 May 2016

Response to Referees' Comments:

We thank three Referees for their helpful and constructive comments. We have made substantial improvements to the paper in responding to each comment. The reviewer comments are shown below in italics, followed by our responses to each point in blue.

***Anonymous Referee #1***

*This is a valuable analysis on an important topic that takes advantage of a detailed model ensemble to address air pollution mortalities. However, the work has a few major deficiencies which ought to be addressed in a revision, as well as several minor comments.*

*1) The paper falls short in putting the work in the context of previous efforts. It cites only two previous papers assessing global health-related impacts of future air pollutants, while there is in fact a larger literature, including both global and regional effects. By not putting their work in the context of the previous work, the authors overstate the novelty of their contribution, and are not able to discuss their work in the context of what is known or unknown in this field.*

We have expanded the number of studies we reference to include more global studies and now include regional studies. While there have been many studies to assess air pollution health effects, the number of global studies that explore future scenarios remains limited. The revised text (p. 2, line 27, to p. 3 line 2) is:

"Previous studies have estimated the present-day global burden of disease due to exposure to ambient ozone and/or PM2.5 (e.g., Apte et al., 2015; Evans et al., 2013; Forouzanfar et al., 2015), with several studies estimating this burden using only output of global atmospheric models (Anenberg et al., 2010; Fang et al., 2013a; Lelieveld et al., 2013; Rao et al., 2012; Silva et al., 2013). However, few studies have evaluated how the global burden might change in future scenarios (Lelieveld et al., 2015; Likhvar et al., 2015; West et al., 2007). Other global studies have estimated future air pollution-related mortality as a by-product of analyses of other future changes, such as the effects of climate change or of climate change mitigation (e.g., Fang et al., 2013b; Selin et al., 2009; West et al., 2013), but do not focus on the range of plausible future mortality as their main purpose. Similarly, studies at local and regional scales have evaluated the mortality impact of changes in air quality due to future climate change (Bell et al., 2007; Chang et al., 2010; Fann et al., 2015; Heal et al., 2012; Jackson et al., 2010; Knowlton et al., 2004, 2008; Orru et al., 2013; Post et al., 2012; Sheffield et al., 2011; Tagaris et al., 2009) but few such studies have evaluated changes beyond 2050."

*2) The paper does not fully take advantage of the potential utility of using a multi-model ensemble in the analysis. As is, it presents straightforward calculations of resulting mortalities, which (for the most part) are very predictable based on the air quality results presented in previous work. The authors do have some very interesting results about the relative contributions of different assumptions of air pollution concentrations, exposure-response functions, population, etc. While some of these results are noted, in my opinion, they are the most interesting implications of this analysis and could be*

*highlighted. However, the authors fall short in this area by not correctly characterizing the uncertainties and variabilities captured by their use of the RCP scenarios (a significant limitation which could be more thoroughly discussed and caveated) and the ACCMIP effort. These ought to be discussed more carefully.*

Taking into account the referee's comments, we have revised the Abstract, Introduction and Discussion sections:

Abstract, (p. 2, line 17): "Mortality estimates differ among chemistry-climate models due to differences in simulated pollutant concentrations, which is the greatest contributor to overall mortality uncertainty for most cases assessed here, supporting the use of model ensembles to characterize uncertainty. Increases in exposed population and baseline mortality rates of respiratory diseases magnify the impact on premature mortality of changes in future air pollutant concentrations and explain why the future global mortality burden of air pollution can exceed the current burden, even where air pollutant concentrations decrease."

Introduction (p. 3, lines 33-37): "All RCPs assume increasingly stringent air pollution controls as countries develop economically, leading to decreases in air pollutant emissions that reflect the different methods of the different RCP groups (e.g., Smith et al., 2011). But as assumptions are similar among the RCPs, the four scenarios do not span the range of possible futures published in the literature for short-term species. For example, other studies have simulated scenarios in which air pollution controls are kept at current levels while underlying trends (e.g., energy use) increase overall emissions (Lelieveld et al. 2015; Likhvar et al. 2015)."

Discussion (p. 12, lines 7-17): "The importance of conducting health impact assessments with air pollutant concentrations from model ensembles, instead of from single models, is highlighted by the differences in sign of the change in mortality among models, and by the marked impact of the spread of model results on overall uncertainty in our mortality estimates. In most cases assessed here (ozone mortality in 2030 relative to 2000, PM2.5 mortality in 2030, 2050 and 2100 relative to 2000), uncertainty in modeled air pollutant concentrations is the greatest contributor to uncertainty in mortality estimates. The differences in air pollutant concentrations reported by the ACCMIP models reflect different treatments of atmospheric dynamics and chemistry, chemistry-climate interactions, and natural emissions in each model (Young et al., 2013). Although there is likely a bias in estimating health effects using air pollutant concentrations from coarse resolution models (Li et al., 2015; Punger and West, 2013), particularly for PM2.5, we do not expect resolution to be an important factor for the differences in simulated concentrations across these coarse resolution global models."

Discussion (p. 13, lines 1-11): "Our results are limited by the range of air pollutant emissions projected by the RCPs, which assume that economic growth strengthens efforts to reduce air pollutant emissions. All RCPs project reductions in anthropogenic precursor emissions associated with more extensive air quality legislation as incomes rise, except for methane in RCP8.5 and for ammonia in all scenarios. These scenarios together do not encompass the range of plausible air pollution futures for the 21st century, as the RCPs were not designed for this purpose (van Vuuren et al., 2011a). Other plausible scenarios have been considered, such as the Current Legislation Emissions and Maximum Feasible Reductions scenarios used by Likhvar et al. (2015) and the Business-As-Usual scenario of Lelieveld et al (2015). As noted above, our global burden estimates for 2050 are considerably lower than the Business-As-Usual scenario of Lelieveld et al. (2015). If economic growth does not lead to stricter air pollution

control, emissions and health effects may rise considerably, particularly for scenarios of high population growth in developing countries (Amman et al., 2013)."

*3) The mortality numbers, while interesting, are not put in proper context such that the reader can understand what they mean. To address this, some comparison with existing literature could be very useful.*

To provide context for our estimates of the future global burden of air pollution on mortality, we estimated the present-day burden and compared with existing literature (Forouzanfar et al. 2015, Silva et al. 2013). Please see Results section, p. 10, lines 28-37):

"For context, we estimate the present-day global burden, using 2000 concentrations, population from Landscan 2011 Population Dataset, and baseline mortality rates from GBD2010, to be 382,000 (121,000 to 728,000) ozone deaths/year and 1.70 (1.30 to 2.10) million PM2.5 deaths/year. These estimates are 18.7% lower for ozone-related mortality and 19.1% lower for PM2.5 mortality than those obtained in our previous study (Silva et al., 2013), reflecting: a) more restrictive mortality outcomes (chronic respiratory diseases rather than all respiratory diseases, and IHD+STROKE+COPD rather than all cardiopulmonary diseases); b) updated population and baseline mortality rates; c) the use of the recent IER model (Burnett et al., 2014) for PM2.5 (instead of Krewski et al., 2009). Compared with the GBD 2013 (Forouzanfar et al. 2015), our estimates are 76% higher for ozone-related mortality and 42% lower for PM2.5-related mortality, likely due to the fact that we estimate the global mortality burden using 1850 concentrations as baseline, while Forouzanfar et al. (2015) consider counterfactual concentrations (theoretical minimum-risk exposure) that are mostly higher for ozone (uniform distribution between 33.3 and 41.9 ppb) and lower for PM2.5 (uniform distribution between 5.9 and 8.7 μg/m3) than 1850 concentrations. In addition, we consider ozone mortality from all chronic respiratory diseases while Forouzanfar et al. (2015) only account for COPD, and we restrict our mortality estimates to adult population while Forouzanfar et al. (2015) include PM2.5 mortality from lower respiratory tract infections in children under 5 years old."

Additionally, we compare our estimates of the global burden of PM2.5 mortality in 2050 with those reported by Lelieveld et al. 2015. While other studies of future air pollution mortality exist, large differences in scenarios and methods make a comprehensive comparison difficult. Please see Results section, p. 11, lines 21-25:

"Our estimates for the global burden of PM2.5 mortality in 2050 (between 1.82 and 3.50 million deaths/year for the four RCPs) are considerably lower than those of Lelieveld et al. (2015) (5.87 million deaths / year for IHD+STROKE+COPD+LC), likely due to the assumption in the RCP scenarios of further regulations on air pollutants, while the Business-As-Usual scenario of Lelieveld et al. (2015) does not assume regulations beyond those currently defined."

*In addition to these major comments, there are several areas in which analyses are not fully described, and/or relevant methods-related information is missing. These are noted below.*

*Minor comments follow:*
*p3, line 30-31: "few studies have evaluated how the global burden might change in future scenarios" and this seems like a small slice of the literature. There are other*

*papers that could be cited here.*

As mentioned above, we have revised the paragraph to include other literature (p. 2, line 27, to p. 3, line 2):
"Previous studies have estimated the present-day global burden of disease due to exposure to ambient ozone and/or PM2.5 (e.g., Apte et al., 2015; Evans et al., 2013; Forouzanfar et al., 2015), with several studies estimating this burden using only output of global atmospheric models (Anenberg et al., 2010; Fang et al., 2013a; Lelieveld et al., 2013; Rao et al., 2012; Silva et al., 2013). However, few studies have evaluated how the global burden might change in future scenarios (Lelieveld et al., 2015; Likhvar et al., 2015; West et al., 2007). Other global studies have estimated future air pollution-related mortality as a by-product of analyses of other future changes, such as the effects of climate change or climate change mitigation (e.g., Fang et al., 2013b; Selin et al., 2009; West et al., 2013), but do not focus on the range of plausible future mortality as their main purpose. Similarly, studies at local and regional scales have evaluated the mortality impact of changes in air quality due to future climate change (Bell et al., 2007; Chang et al., 2010; Fann et al., 2015; Heal et al., 2012; Jackson et al., 2010; Knowlton et al., 2004, 2008; Orru et al., 2013; Post et al., 2012; Sheffield et al., 2011; Tagaris et al., 2009) but few such studies have evaluated changes beyond 2050."

*p3, line 26-27: "RCPs… do not span the range of possible futures published in the literature for short-term species." This is a key point and it could be highlighted.*

We have strengthened the discussion of this point in responding to major comment #2 above.

"All RCPs assume increasingly stringent air pollution controls as countries develop economically, leading to decreases in air pollutant emissions that reflect the different methods of the different RCP groups (e.g., Smith et al., 2011). But as assumptions are similar among the RCPs, the four scenarios do not span the range of possible futures published in the literature for short-term species. For example, some other studies have simulated scenarios in which air pollution controls are kept at current levels while underlying trends (e.g., energy use) increase overall emissions (Lelieveld et al. 2015; Likhvar et al. 2015)."

*p4, line 4-6: but the ACCMIP is coarser. The mortality estimates thus should be justified. Also, line 33-34 on same page: this regridding to a scale finer than that modeled should be better described and justified.*

Here we regrid each model to a much finer resolution (0.5°x0.5°). We select a resolution that is much finer than any model to limit errors associated with regridding. For each individual model, the fact that the results were regridded to a finer resolution should not influence the results. For the multi-model ensemble, however, our regridding takes maximum advantage of how the different model grids line up or overlay one another. This is preferable to regridding to a common coarse resolution grid, as some of the information of how grids overlay on one another would be lost. We have used these methods previously for the same reason (Anenberg et al., 2009, 2014; Silva et al., 2013).

Our 0.5°x0.5° gridded estimates do not truly represent the fine-scale structure of air pollutant concentrations as a model simulation at this resolution might be able to achieve, since no model

was run at this fine resolution.  As a result, we clearly indicate in the paper that the resolution is insufficient to capture local or urban scale effects (p.4, lines 16-18):

"Mortality estimates are obtained at a sufficiently fine horizontal resolution (0.5°x0.5°) to capture both global and regional effects and inform regional and national air quality and climate change policy, but are not expected to capture local scale (e.g., urban) air pollution effects."

Also, we have revised the text as follows (p. 5, lines 8-11)

"The native grid resolutions of the 14 models varied from 1.9°x1.2° to 5°x5°; we regrid ozone and PM2.5 species surface concentrations from each model to a common 0.5°x0.5° horizontal grid to take maximum advantage of how the grids of different models overlap, following Anenberg et al. (2009, 2014) and Silva et al. (2013)."

And we have added discussion of the uncertainty in results brought about by the coarse resolution of global models (p. 12, lines 6-10):

"The differences in air pollutant concentrations reported by the ACCMIP models reflect different treatments of atmospheric dynamics and chemistry, chemistry-climate interactions, and natural emissions in each model (e.g., Young et al., 2013). Although there is likely a bias in estimating health effects using air pollutant concentrations from coarse resolution models (Li et al., 2015; Punger and West, 2013), particularly for PM2.5, we do not expect resolution to be an important factor for the differences in simulated concentrations across coarse resolution models."

*p5, line 19-20: "similar to Silva et al 2013...except for..." Does this mean exactly the same as the Silva et al 2013 paper except for those two differences? The description is unclear, and the language here could be more precise.*

The methods are identical except for those two differences. As we detail later in the Methods section, "we apply the IER model instead of RRs from Krewski et al. (2009), used by Silva et al. (2013), as the newer model should better represent the risk of exposure to PM2.5, particularly at locations with high ambient concentrations", and we use projections of population and baseline mortality rates to estimate the effect of future air pollution "considering the population that will potentially be exposed to those effects."

We have revised the initial sentence of the Methods section to make it more precise (p. 5, lines 26-30):

"We estimate future air pollution-related cause-specific premature mortality using generally the same methods as those used by Silva et al. (2013) to obtain present-day estimates, but with two important differences: (1) we use the recently published Integrated Exposure-Response (IER) model for PM2.5 (Burnett et al., 2014), and (2) we use projections of population and baseline mortality rates from the International Futures (IFs) integrated modeling system (Hughes et al., 2011)."

*p6, line 9-10: using a common projection of population across the RCPs introduces both consistency in this analysis, but inconsistency relative to underlying social drivers. The implications of this choice should be discussed further, with quantifications of the*

*magnitude as well as the direction.*

Taking into account the referee's comments, we have revised the text (p. 6, lines 32-33):

"Population projections from IFs differ from those underlying each RCP, but lie within the range of the RCPs (Figure S4). In 2030, global total population in IFs is within 0.08% of that reported for RCP2.6, RCP4.5 and RCP6.0 and 5% lower than for RCP8.5; however, in 2100 IFs projects larger global populations than RCP2.6 (+7%), RCP4.5 (+13%) and RCP6.0 (+2%) and considerably lower than RCP8.5 (-27%). IFs projects rising baseline mortality rates for cardiovascular diseases (CVD) and RESP, globally and in most regions (particularly in East Asia and India), reflecting an aging population. By using projections from IFs, we have a single source of population and baseline mortality rates, assuring their consistency and enabling us to isolate the effect of changes in air pollutant concentrations across the RCPs. Had we used the population projections from each scenario, the magnitude of the changes (increases or decreases in premature mortality relative to 2000) would likely increase in RCP8.5, but decrease in RCP2.6, RCP4.5 and RCP6.0."

*p6, line 27-28: I can guess what the authors are referring to here, but the language could be easily misinterpreted (as the authors do actually look at the influence of climate on air pollutants themselves, just not modifications in ER factors). Rephrase?*

We have revised the sentence, as suggested (p.7, lines 9-12):

"Our results do not reflect the potential synergistic effect of a warmer climate on air pollution-related mortality, i.e., we do not account for potential changes in the exposure-response relationships at higher temperatures (Pattenden et al. 2010; Wilson et al., 2014 and references therein)."

p7, line 1-5: are potential correlations between different RRs accounted for in the Monte Carlo sampling? If so, how is that done? If not, the spread could be artificially narrowed. Please discuss.

It is not clear that there would be correlations between the RRs for different causes of death resulting from PM2.5, or if there are, it is not clear how they would be modeled.  We have evaluated uncertainty for each cause of death separately and then added these results together. For ozone, there is only one cause of death and this is not an issue.  For PM2.5, there are four causes of death.  The Referee is correct that if there are correlations between these RRs, our methods would underestimate the overall uncertainty for PM2.5.

We have added a sentence to acknowledge this limitation (p. 7, lines 29-31):

"Uncertainty from the RRs is propagated separately for each model-scenario-year to mortality estimates in each grid cell, through 1000 Monte Carlo (MC) simulations, i.e. we repeat the calculations in each grid cell 1000 times using random sampling of the RR variable. For ozone, we use the reported 95% Confidence Intervals (CIs) for RR (Jerrett et al., 2009) and assume a normal distribution, while for PM2.5 we use the parameter values reported by Burnett et al. (2014) for 1000 MC simulations (GHDx 2013). Then for each of the 1000 simulations, we add mortality over many grid cells to obtain regional and global mortality and estimate the empirical

mean and 95% CI of the regional and global mortality results. We assume no correlation between the RRs for the four causes of death; thus we may underestimate the overall uncertainty for PM2.5 mortality estimates."

*p7, line 10: for the ACP audience, please describe 'tornado analysis' more thoroughly and quantitatively. Also, it is not addressed again, and there is no associated figure that corresponds to a traditional tornado-type plot.*

We have revised the sentence in Methods to include a description of the tornado analysis (p. 7, line 34, to p. 8 line 2):

"We also estimate the contribution of uncertainties in RR and in air pollutant concentrations to the overall uncertainty in mortality estimates using a tornado analysis; we obtained global mortality estimates treating each variable as uncertain individually (year 2000 concentrations, future year concentrations, RR for ozone, and the four parameters in the IER model for PM2.5) and used central estimates for all other variables, and then calculated the contribution of each variable to the overall uncertainty (when all variables are treated as uncertain simultaneously)."

The quantitative results from the tornado analysis are included in the following sentences:

(p. 9, lines 10-13) [ozone] "While uncertainty in RR and in modeled ozone concentrations have similar contributions to overall uncertainty in mortality results in 2050 (51% and 49%, respectively), in 2030 modeled ozone concentrations are the greatest contributor (81%), and in 2100 uncertainty in RR contributes the most to overall uncertainty (88%)."

(p. 10, lines 12-16) [$PM_{2.5}$] "Uncertainty in modeled PM2.5 concentrations in 2000 is the greatest contributor to overall uncertainty (59% in 2030, 45% in 2050, and 49% in 2100), followed by uncertainty in modeled PM2.5 in future years (40% in 2030, 26% in 2050 and 32% in 2100). Uncertainty in RR has a negligible contribution to overall uncertainty in 2030 (<1%), as the multi-model mean mortality change happens to be near zero (one model projects a large increase while the other five models project decreases), but contributes 29% in 2050 and 20% in 2100."

We do not show a traditional tornado plot since there are few variables treated as uncertain, and we combine related uncertainties together (the IER parameters). The full results of such a plot are communicated in the above sentences.

*p7. line 7: While the authors do have a certain spread of air pollutant concentrations, this should not be taken as a measure of 'uncertainty'. It is decidedly not a quantitative uncertainty analysis, as there are many other factors affecting 'uncertainty' in air pollutant concentrations that are not captured by the ACCMIP ensemble. This should be noted and discussed, and language carefully examined throughout the paper.*

We consider that the spread of air pollutant concentrations across models is a measure of uncertainty in air pollutant concentrations, although it does not account for uncertainty in emission inventories or for potential bias in modeled air pollutant concentrations. We have revised the Methods and the Discussion sections to address the referee's comments:

(p. 7, lines 31-34)
"Uncertainty in air pollutant concentrations is based on the spread of model results by calculating the average and 95% CI for the pooled results of the 1000 MC simulations for each model. This estimate of uncertainty in concentrations does not account for uncertainty in emissions inventories (as the ensemble used identical emissions) or for potential bias in modeled air pollutant concentrations."

(p. 12, lines 29-37)
"The spread of model results does not account for uncertainty in emissions inventories, as all ACCMIP models used the same projections of anthropogenic emissions. Moreover, climate and air quality interactions and feedbacks are sufficiently understood to be fully reflected in modeled air pollutant concentrations, and global models simplify atmospheric physics and chemical processes. This is particularly important when modeling air quality given scenarios of future emissions and climate change. For example, most global models do not fully address climate sensitivity to biogenic emissions (e.g. isoprene, soil NOx and methane) and stratosphere-troposphere interactions (e.g. stratospheric influx of ozone). A better understanding of aerosol-cloud interactions, of the impact of climate change on wildfires, and of the impact of land use changes on regional climate and air pollution is also crucial."

*p 7, line 19: "In some cases…" This sentence is confusing. Rephrase?*

We have revised the sentence as suggested:

(p. 8, lines 10-13) "In some cases, the changes in future mortality due to changes in future concentrations relative to 2000 show a different trend than the global mortality burden; this difference reflects the combined effects of future changes in concentrations relative to 1850, exposed population and baseline mortality rates."

*p 9, lines 21-22: I'm not clear what was done here. This should be addressed in detail in methods.*

These results reflect the following text in Methods (p. 5, lines 6-8):

"We use our PM2.5 estimates to obtain all mortality results, and perform a sensitivity analysis using the PM2.5 concentrations reported by four models using their own PM2.5 formulas, which differed among models, as reported in Silva et al. (2013)."

We revised the text in Results to expand the explanation (p. 10, lines 17-18):

"We compared mortality results using our estimates of PM2.5 from the sum of reported species with results using PM2.5 reported by four models applying their own formula to estimate PM2.5 (Figure 7)."

*p 10, line 31-32: This difference is noted. However, anyone familiar with the ACCMIP effort could have gleaned this simply from the previous reported results. What is new here? Why is this particularly significant in terms of mortality?*

We agree that there is a spread of results among the ACCMIP models. But it is not entirely obvious how this would influence the spread of mortality results, since one would have to account for the uneven distribution of population around the world, as we have done here. We highlight here the results specifically for estimates of human mortality, and show in the next sentence how the uncertainty contributed by the spread of model results compares with the uncertainty in the concentration response function itself (p. 12 , lines 8-13):

"The importance of conducting health impact assessments with air pollutant concentrations from model ensembles, instead of from single models, is highlighted by the differences in sign of the change in mortality among models, and by the marked impact of the spread of model results on overall uncertainty in our mortality estimates. In most cases assessed here (ozone mortality in 2030 relative to 2000, PM2.5 mortality in 2030, 2050 and 2100 relative to 2000), uncertainty in modeled air pollutant concentrations is the greatest contributor to uncertainty in mortality estimates."

*p 11, line 16+ This could be discussed in more depth, including more quantitatively, as it's a key limitation of the authors' analysis.*

We have revised the text to account for this comment, as well as in responding to major comment #2 above. We highlight the comparison with results from other studies using different scenarios:

Discussion (p. 13, line 1-11): "Our results are limited by the range of air pollutant emissions projected by the RCPs, which assume that economic growth strengthens efforts to reduce air pollutant emissions. All RCPs consider reductions in anthropogenic precursor emissions associated with more extensive air quality legislation as incomes rise, except for methane in RCP8.5 and for ammonia in all scenarios. These scenarios together do not encompass the range of plausible air pollution futures for the 21st century, as the RCPs were not designed for this purpose (van Vuuren et al., 2011a). Other plausible scenarios have been considered, such as the Current Legislation Emissions and Maximum Feasible Reductions scenarios used by Likhvar et al. (2015) and the Business-As-Usual scenario of Lelieveld et al (2015). As noted above, our global burden estimates for 2050 are considerably lower than the Business-As-Usual scenario of Lelieveld et al. (2015). If economic growth does not lead to stricter air pollution control, emissions and health effects may rise considerably, particularly for scenarios of high population growth in developing countries."

**Anonymous Referee #2**

*This manuscript uses the RCPs to project estimated air pollutant levels and health impacts globally for 10-year intervals between 2000 and 2100. It advances previous publications through the use of projected baseline mortality and population size along with projected air pollutant concentrations and therefore one can isolate the impacts of projected emissions from those of demographic changes in estimating future health impacts from air pollution. Further, the use of ensemble forecasts allows for the evaluation of the role of model variability in future estimates. Interestingly, while mortality impacts related to PM2.5 levels are projected to decrease under all scenarios, mortality from ozone exposure is projected to increase in all scenarios due to changes in population demographics, the absence of widespread decreases in ozone*

*concentrations, increases in methane and climate warming. There are two main analyses in the manuscript:1) the impact of concentration changes relative to those in the year 2000 which is focused on the effects of future emissions and the variability between the different models, and 2)the assessment of the overall burden of disease attributable to air pollution in future years relative to pre-industrial (1850) concentrations where the relative impacts of emissions, and population projections are compared (cases A and B).*

*General comments*
*Overall the manuscript provides unique new information to assess both potential future health impacts under well-defined scenarios and the role of model variability, uncertainty in concentration-response functions, uncertainty in emissions and the role of demographic changes in the estimation of future impacts. While the absolute numbers from the simulations are interesting, arguably more important is the assessment of uncertainty and the relative roles of different factors (demographics, emissions) in future estimates. For this component of the manuscript, decreasing the emphasis on the absolute numbers while providing more relative comparisons would help the reader sort through all of the results. Further the manuscript would benefit from some clear take-home messages on the relative impacts of future emissions and demographic changes and on the largest contributors to overall uncertainty. This information is in the manuscript but is hard to find and needs to be brought forward (even if it means decreasing emphasis on the absolute numbers).*

We thank Referee #2 for these encouraging and helpful comments. We have made changes throughout the manuscript, particularly in the Discussion and Conclusions sections to decrease emphasis on particular numerical results, and to strengthen our communication of key messages, and have responded to the specific comments below.

*The estimates for 2000 are low compared with other similar estimates and the authors attribute this to the choice of counterfactual. Given that the counterfactual is a choice, it would seem useful to isolate the impact of the choice of counterfactuals if the absolute number is being emphasized – some simple sensitivity analyses in which, for example, the Global Burden of Disease counterfactuals were applied, would be useful.*

Following the referee's suggestion, we included a simple sensitivity analysis considering the global burden for 2000 using the GBD counterfactuals. We have improved the comparison with GBD 2013 and added the comparison with estimates using the GBD counterfactuals to the Results section (p. 10, line 34, to p. 11, line 10):

"Compared with the GBD 2013 (Forouzanfar et al. 2015), our estimates are 76% higher for ozone-related mortality and 42% lower for PM2.5-related mortality, likely due to the fact that we estimate the global mortality burden using 1850 concentrations as baseline, while Forouzanfar et al. (2015) consider counterfactual concentrations (theoretical minimum-risk exposure) that are mostly higher for ozone (uniform distribution between 33.3 and 41.9 ppb) and lower for PM2.5 (uniform distribution between 5.9 and 8.7 µg/m3) than 1850 concentrations.  In addition, we consider ozone mortality from all chronic respiratory diseases while Forouzanfar et al. (2015) only account for COPD, and we restrict our mortality estimates to adult population while Forouzanfar et al. (2015) include PM2.5 mortality from lower respiratory tract infections in young children. As a sensitivity analysis, when we apply a counterfactual of 33.3 ppb (instead of using 1850 concentrations), our ozone-related mortality estimates are 23% higher for the multi-model mean, varying between +10% and +52% among models. Similarly, using the IER model

counterfactual, our PM2.5-related mortality estimates are 22% lower for the multi-model mean, varying between -8% and -44% among models."

*Future ozone and PM2.5 attributable mortality is clearly driven by China and India; given this it might be useful to present (or at least comment on) the model variability in these regions as what appears to be overall agreement across most of the models may be a result of smoothing due to other regions which have relatively minor impacts on future trends.*

We have added to the Supplemental Material maps of the coefficient of variation (Figures S8 and S9) to show the spatial distribution of model variability for all RCPs and all future years. In most cases, PM2.5 concentrations show lower variability in India and China than in other regions across RCPs and future years. In most cases, variability of ozone concentrations across models is much greater in 2030 than in 2050 and 2100, including in China and India. We have added text to the Results section to address this point (p. 9, line 36 to p. 10, line 2):

"East and South Asia are the regions with the greatest projected mortality burdens, and the variability in PM2.5 among models is typically less in these regions than in several other regions globally, depending upon the scenario and year (Figure S9)."

*Specific comments Abstract should be more consistent in presenting uncertainty in estimates and should include some quantification of uncertainty. Abstract should also provide more emphasis on uncertainty and relative impacts of different sources for the burden of disease estimates*

We have revised the Abstract taking into account the reviewer's comment (p. 2, lines 8, 12-13 and 17):

"However, the global mortality burden of ozone markedly increases from 382,000 (121,000 to 728,000) deaths/year in 2000 to between 1.09 and 2.36 million deaths/year in 2100, across RCPs, mostly due to the effect of increases in population and baseline mortality rates. PM2.5 concentrations decrease relative to 2000 in all scenarios, due to projected reductions in emissions, and are associated with avoided premature mortality, particularly in 2100: between -2.39 and -1.31 million deaths/year for the four RCPs. The global mortality burden of PM2.5 is estimated to decrease from 1.70 (1.30 to 2.10) million deaths/year in 2000 to between 0.95 and 1.55 million deaths/year in 2100 for the four RCPs, due to the combined effect of decreases in PM2.5 concentrations and changes in population and baseline mortality rates. Trends in future air pollution-related mortality vary regionally across scenarios, reflecting assumptions for economic growth and air pollution control specific to each RCP and region. Mortality estimates differ among chemistry-climate models due to differences in simulated pollutant concentrations, which is the greatest contributor to overall mortality uncertainty for most cases assessed here, supporting the use of model ensembles to characterize uncertainty. Increases in exposed population and baseline mortality rates of respiratory diseases magnify the impact on premature mortality of changes in future air pollutant concentrations and explain why the future global mortality burden of air pollution can exceed the current burden, even where air pollutant concentrations decrease."

As we did not estimate uncertainty for future scenarios (except for RCP8.5), we do not report uncertainty ranges for future results in the abstract.

*L89 -Lim et al should be updated with Forouzanfar et al., 2015*

We have updated the reference as suggested (p. 2, line 29; p. 10, line 34)

*L102 - suggest that in future ozone concentrations will decrease with climate change; can this be reconciled with observations on global increases during recent periods? (Emissions vs warming?)*

As we state in the paper, concentrations of ozone are expected to increase in polluted regions in the warm season, as a result of future climate change. Ozone is likely to decrease in remote regions as a result of climate change.  Our previous work analyzed the effects of past climate change on air pollution-related mortality, finding a small influence (Silva et al., 2013). This finding is consistent with the current literature, which reports that the effect of past emissions changes far outweighs the effect of climate change at present. As we do not focus on climate change in this paper, we have not changed the text to address this point.

*L239 How do IF projections compare with current numbers, i.e. from the Global Burden of Disease ( ̃for 2010)?*

IF projections for 2010 and GBD 2010 estimates of age-standardized mortality rates (deaths per 100,000 people) are:

| Diseases | IF | GBD 2010 |
|---|---|---|
| Cardiovascular | 234.9 | 234.8 |
| Chronic Respiratory | 58.4 | 57.0 |
| Neoplasms | 106.9 | 121.4 |

We have added this table to the Supplemental Material (Table S4).

Data in Figure S5 are for adult population only.  We added this text to the paper (p. 7, lines 19-20):

"IFs projections for 2010 are comparable to GBD 2010 estimates (Lozano et al., 2012) for CVD (+0.04%), RESP (+2.5%) and neoplasms (-12%)."

*L283 –Should mention in limitation/discussion that the absence of uncertainty in the IF projections may be as important as other sources of uncertainty and that this uncertainty would increase over time (i.e. 2100 vs 2030)*

We have revised the text in the Discussion section taking into account this comment (p. 12, lines 26-28):

"Uncertainty is evaluated for a single future population projection, not accounting for the wide range of projections in the literature, and does not reflect uncertainty in baseline mortality rates, as these are not reported; uncertainties in both population and baseline mortality rates would be expected to increase with time into the future."

*L299-310 –There would appear to be  ~20x variability estimates for the different RCP scenarios - this is very large and clearly makes the case that emissions DO matter - it seems that this point should also be brought out a bit more.*

This is a good point.  We have revised the conclusions section to make this point in a more prominent place in the manuscript (p. 13, lines 19-22):

"These reductions in ambient air pollution-related mortality reflect the decline in pollutant emissions projected in the RCPs, but the large range of results from the four RCPs highlights the importance of future air pollutant emissions for ambient air quality and global health."

The theme of the importance of emissions is again reiterated in the last paragraph of the paper.

*L404 what are the 1850 concentrations that are used as the counterfactual? These should be provided in the text.*

We do not use a single value for 1850 concentrations that applies globally.  Rather, a different value is present in each grid cell as a result of the model simulations.  We have added maps of 1850 concentrations for ozone and PM2.5 to the supporting information (Figures S4 and S5).

*Apte JS et al., ES&T 2015 also estimates future mortality assuming only changes on population – it would be useful to cite this paper and make some rough comparisons*

We have included Apte et al. 2015 in the literature cited in the Introduction (p. 2, lines 28)

"Previous studies have estimated the present-day global burden of disease due to exposure to ambient ozone and/or PM2.5 (e.g., Apte et al., 2015; Evans et al., 2013; Forouzanfar et al. 2015Lim et al., 2012), with several studies estimating this burden using only output of global atmospheric models (Anenberg et al., 2010; Fang et al., 2013a; Lelieveld et al., 2013; Rao et al., 2012, ; Silva et al., 2013). However, few studies have evaluated how the global burden might change in future scenarios (e.g. Lelieveld et al., 2015; Likhvar et al., 2015; West et al., 2007)."

and in the comparisons included in the Results (p. 11, lines 17-24):

"For PM2.5, the increase in exposed population and the decline in concentrations have a much greater effect than changes in baseline mortality rates (Figure 9). These results are similar to those of Apte et al. (2015) who report a stronger effect of projected demographic trends in India and China in 2030 than of changes in baseline mortality rates. Our estimates for the global burden of PM2.5 mortality in 2050 (between 1.82 and 3.50 million deaths/year for the four RCPs) are considerably lower than those of Lelieveld et al. (2015) (5.87 million deaths / year for IHD+STROKE+COPD+LC), likely due to the assumption in the RCP scenarios of further regulations on air pollutants, while the Business-As-Usual scenario of Lelieveld et al. (2015) does not assume regulations beyond those currently defined."

L472 "preature" typo

Corrected.

**Anonymous Referee #3**

*General comments: This study performs a global health impact assessment from ambient air pollution, using chemical transport or chemistry-climate models, for a set of RCP scenarios, for the years 2000, 2030, 2050 and 2100. Similar studies have been published before (properly acknowledged by the authors). The novelty of this study lies in the use of an ensemble of models, allowing for an evaluation of the contribution of model-calculated population exposure to pollution in the total uncertainty on the health impact. However a comparison of the outcome with previous studies, both for present day and future projections, is not obvious because of differences in methodology.*

We thank Referee #3 for the constructive comments and have addressed the specific comments below. We have improved our presentation of previous studies in the Introduction, Although we compare our results for the present-day GBD with several studies, we agree that it is difficult to do a systematic comparison of future mortality with the few comparable studies available, and only compare quantitatively with the results of Lelieveld et al. (2015).

*Specific comments: In the paper two ways are used to evaluate the impact of emission scenarios for the future on human health: 1) By using future demographics and health statistics, and combining these with exposure to year 2000 pollutant levels and to pollutant levels corresponding to projected emissions for the specific year respectively and making the difference 2) by calculating the absolute number of mortalities for each considered year and making the difference with mortalities for 1850 ('mortality burden'). It took me a while to understand that reported 'avoided' and 'excess' mortalities refer to method 1). It should be better explained in the methodology section. Usually, avoided or excess mortalities for a given scenario are calculated versus a reference scenario for the same year (e.g. a stringent policy versus a business-as-usual as reference case). It's not clear here what the year 2000 pollution transposed to 2030 and 2050 actually represents as a reference. The avoided or excess mortalities can not be directly linked to specific policies (which pathway would have led to the year 2000 levels in 2030 - 2050 - 2100?). Wouldn't it make more sense to use e.g. RCP 8.5 as a reference, and evaluate the benefits of the 2.6 and 4.5 pathways? Using year 2000 pollution levels as a reference for future years also introduces an issue with exposure; concentration field spatial distribution is linked to population spatial patterns – in particular for PM. Does is make sense to overlay year 2000 pollution spatial patterns with year xxxx population spatial distribution?*

We use year 2000 pollution levels as a common counterfactual against which future pollutant concentrations are evaluated. In doing so, we evaluate future concentrations as they affect future population, relative to the year 2000 concentrations. That is, we evaluate air pollution mortality in future years relative to the case where that future population breathes air from the year 2000. This approach is analogous to using 1850 concentrations as a counterfactual, and we choose to use 2000 rather than 1850 as the main results that we present for the reasons we state in the paper – particularly because it does not require assumptions about the shape of the

concentration-response function at the very low concentrations present in 1850. We also prefer comparing with 2000 concentrations because that is the state of air pollution with which we are all familiar.

We have revised the text in the Methods section to improve discussion of these points, and to clarify the use of 'avoided/excess' mortality (p. 5, line 35 to p. 6, line 1):

"We calculate changes in premature mortality by applying the change in pollutant concentrations in each future year (2030, 2050, 2100) relative to year 2000 concentrations - the present-day state of air pollution - to the future population. We therefore estimate 'avoided' or 'excess' premature mortality due to decreases or increases in air pollutant concentrations in the future years relative to 2000 concentrations. This approach differs from a calculation of the global burden of air pollution-related mortality since we use 2000 rather than 1850 concentrations as baseline. We estimate mortality changes due to future concentration changes, relative to the present, to avoid applying the health impact function at very low concentrations where there is less confidence in the exposure-response relationship. For example, the simulated 1850 air pollutant concentrations are often below the lowest measured value of the American Cancer Society study (Jerrett et al., 2009; Krewski et al., 2009). For illustration, we also estimate mortality relative to 1850 concentrations, which could be regarded as global burden of disease calculations, following Silva et al. (2013)."

*Mortalities are estimated at 0.5x0.5 deg resolution: is this just a regridding of the native model resolution or was any downscaling done to better estimate the exposure indensely populated areas? Apparently the concentrations are just regridded; this cannot be considered as a proper population-weighted exposure estimate at the coarse resolution of the models, as all population within a single grid will be exposed to the same level.*

We have regridded the concentration fields for each model individually, without doing any downscaling, as the Referee suggests. We regrid to a finer resolution to improve our estimates for the multi-model average, retaining information on how the grids for different models line up. We have improved the text to clarify our purpose in regridding (p. 5, lines 8-11):

"The native grid resolutions of the 14 models varied from 1.9°x1.2° to 5°x5°; we regrid ozone and PM2.5 species surface concentrations from each model to a common 0.5°x0.5° horizontal grid to take maximum advantage of how the grids of different models overlap, following Anenberg et al. (2009, 2014) and Silva et al. (2013)."

*Regarding the use of Burnett's IER functions: specify whether age-specific functions have been used or all-ages. From what is written in the first par. of page 15, I understood that the Burnett functions have been applied without the counterfactual value? In fact it is not well explained how teh difference with 1850 was made: by first subtracting 1850 concentrations and then applying the exposure-response functions, or by applying exposure-response functions to both years and then subtracting mortalities. And how was it done for calculating the excess/avoided mortalities relative to year 2000?*

When applying the IER model, we used age-specific functions when reported by Burnett et al. 2014 (IHD and Stroke). We have revised the text to mention this (p. 6, lines 16-17):

"We also estimate PM2.5-related mortality due to ischemic heart disease (IHD), cerebrovascular disease (STROKE), chronic obstructive pulmonary disease (COPD) and lung cancer (LC), using RRs from the IER model (Burnett et al., 2014). We use RR per age group for IHD and STROKE and RR for all-ages for COPD and LC."

We used the RRs (central estimate) from the IER model reported by Burnett et al. 2014 for $PM_{2.5}$ concentrations up to 300 $\mu g/m^3$ (GHDx 2013) to obtain the deterministic mortality estimates. For the uncertainty analysis, we use the values for parameters alpha, gamma, delta and $z_{cf}$ (counterfactual) reported by Burnett et al. (2014) for 1000 simulations (GHDx 2013).

We have revised the text to make this more clear (p. 7, lines 25-27):

"For ozone, we use the reported 95% Confidence Intervals (CIs) for RR (Jerrett et al., 2009) and assume a normal distribution, while for PM2.5 we use the values for the parameters alpha, gamma, delta and zcf (counterfactual) reported by Burnett et al. (2014) for 1000 MC simulations (GHDx 2013)."

We applied the exposure-response function to both years (future years and 1850 for global burden and futures years and 2000 for the excess/avoided future mortality) and then subtracted the mortality estimates. We have revised the text to explain this (p. 6, lines 1-5):

"To estimate ozone mortality, we apply the exposure-response function to the difference in ozone concentrations, while for PM2.5 mortality we apply the exposure-response function to concentrations in each year (future years and 2000) and then subtract the mortality estimates. We therefore estimate 'avoided' / 'excess' premature mortality due to decreases / increases in air pollutant concentrations in the future years relative to 2000 concentrations."

*The numbers in Table S3 do not seem to be consistent with year 2030 mortalities in Figure 4: In Table S3 only 2 models predict a global mean decrease in PM2.5 for RCP2.6 in 2030. In Figure 4 all models except 1 show a decrease in mortalities by 2030...Similar for the other RCPs; most flagrant for RCP8.5 where all PM2.5 appears to increase globally but only 1 model leads to an increase in mortality. How to explain this?*

The numbers in Table S3 are global averages and are often close to zero in 2030. The spatial distribution of concentrations, how they overlay on baseline mortality rates, and the magnitudes of baseline (2000) and future concentrations (which determine their place in the IER exposure-response curves) have significant impacts on the mortality estimates due the non-linearity in the IER model. We have checked both the calculation of mortality and the calculation of population-weighted concentrations to ensure that both were done correctly.

We thank the Referee for this comment, and think that it is a nice way to illustrate the importance of the nonlinearity of the IER function. We have added text to show this (p. 10, lines 3-8):

"Future PM$_{2.5}$-related mortality estimates are influenced by the nonlinearity of the IER function. For example, in RCP8.5 in 2030, all models project an increase in global population-weighted concentration (Table S3) but all models except one show decreases in global PM$_{2.5}$-related mortality (Figure 4). This outcome results in part because PM$_{2.5}$ increases are projected in regions with high concentrations (particularly East Asia) that are on the flatter part of the IER curve, whereas PM$_{2.5}$ decreases in regions with low concentrations (North America and Europe) have a steeper slope and therefore a greater influence on global mortality."

*Table S4: should be mentioned as 'CHANGE' in mortalities between year 2000 pollution levels and respective scenario/year pollution levels. Also on Page 11, "Global future premature mortality rises from 264,000 (-39,300 to 648,000) deaths in 2030 to 316,000 (-310 187,000 to 1.38 million) deaths in 2100" may cause confusion as these are again changes compared to 2000 pollution levels.*

The caption of Table S4 (now Table S5) has been revised to "Change in global respiratory premature ozone mortality in 2030, 2050 and 2100 for all RCPs (considering the change in future ozone concentrations relative to 2000 concentrations), showing the multi-model average (deaths/year) for RCP2.6, RCP4.5 and RCP6.0 deterministic estimates and the empirical mean with 95% CI in parenthesis for RCP8.5 probabilistic estimates (including uncertainty in the RRs and across models). These results correspond to Figure 1. All numbers are rounded to three significant digits."

The caption of Table S6 (now Table S7) was revised in a similar way.

The text on p. 9 (lines 6-8) has been changed to:
"Global future premature mortality changes from 264,000 (-39,300 to 648,000) deaths in 2030 to 316,000 (-187,000 to 1.38 million) deaths in 2100."

*The fact that the range spans from negative to positive implies that the result is not significantly different from 0?*

We do not include statistical significance testing to evaluate whether results are significantly different from zero. Rather, we present results such that the reader can understand both the results for individual models with uncertainty from the concentration-response function and the net uncertainty when evaluating over all models.

*What has been the benefit of the multi-model analysis? And what can be learned from analyzing the RCP scenarios? Are the outcomes plausible in the light of the implicitly assumed rather stringent pollution controls?*

We have improved the Discussion and Conclusion sections taking into account these comments, as shown, for example, in the following excerpts:

(Discussion: p. 12, lines 1-7):
"In all RCP scenarios but RCP8.5, stringent air pollution controls lead to substantial decreases in ozone concentrations through the 21st century, relative to 2000. For RCP8.5, the higher

baseline GHG (including methane) and air pollutant emissions lead to increases in future ozone concentrations. In contrast, global PM2.5 concentrations show a decreasing trend across all RCP scenarios. These changes in air pollutant concentrations, combined with projected increases in baseline mortality rates for chronic respiratory diseases, drive ozone mortality to become more important relative to PM2.5 mortality over the next century.

The importance of conducting health impact assessments with air pollutant concentrations from model ensembles, instead of from single models, is highlighted by the differences in sign of the change in mortality among models, and by the marked impact of the spread of model results on overall uncertainty in our mortality estimates."

Conclusion (p. 13, lines 19-24):
"These reductions in ambient air pollution-related mortality reflect the decline in most emissions projected in the RCPs, but the large range of results from the four RCPs highlights the importance of future air pollutant emissions for ambient air quality and global health. Mortality estimates differ among models and we find that, for most cases, the contribution to overall uncertainty from uncertainty associated with modeled air pollutant concentrations exceeds that from the RRs."

*The results section is dry and hard to digest with long lists of numbers of mortality changes per scenario, per region, with differences between models – all things that are much easier to read from the figures than in the text. For the reader it is hard to keep an overview and grasp the major message. Suggest to reduce and condense this section to most salient observations that are maybe not directly evident from the figures.*

We made minor changes to the Results section itself - to provide less emphasis on particular numerical results – but have made significant changes to the Discussion and Conclusions sections where the major points are now reiterated more clearly.

*Discussion section: it looks like there is an increasing relative importance of O3 as health impact compared to PM for the future (what is the relative contribution of each pollutant to total pollution mortality burden in each year, each scenario?) – this may be worth a few lines of discussion.*

We have added text in the Discussion and Conclusions sections taking into account this comment (p. 12, lines 1-7, and p. 13, lines 31-34):

"In all RCP scenarios but RCP8.5, stringent air pollution controls lead to substantial decreases in ozone concentrations through the 21st century, relative to 2000. For RCP8.5, the higher baseline GHG (including methane) and air pollutant emissions lead to increases in future ozone concentrations. In contrast, global PM2.5 concentrations show a decreasing trend across all RCP scenarios. These changes in air pollutant concentrations, combined with projected increases in baseline mortality rates for chronic respiratory diseases, drive ozone mortality to become more important relative to PM2.5 mortality over the next century."

"A strong decline in PM2.5 concentrations for all RCPs together with demographic trends in the 21st century (with a projected substantial increase in exposed population) lead to a rising importance of ozone relative to PM2.5 for the global burden of ambient air pollution-related mortality."

*It is surprising that for the same emission scenarios, models have such different outcomes. Does the resolution play a role here? What could be done to improve the exposure estimate? Downscaling techniques? Use of regional models? Is it possible to evaluate the error made by using course resolution models?*

We do not have an easy way to separate the influence of resolution on health outcomes. Previous work that is now cited in the paper (Punger and West, 2013; Li et al., 2015) suggests that there is a bias in estimating health effects from using coarse resolution models that is greater for PM2.5 than for ozone. However, we expect that resolution does not play a big role in the difference between models. That is, there may be a bias from coarse grid resolution relative to fine resolution – and we have added text to acknowledge this point (below) – but the bias caused by resolution from one coarse grid to another should be fairly small. Instead, the difference is caused by differences in modeled concentrations, reflecting the different meteorology and atmospheric chemistry within the different models.

We have added the following sentences to the Discussion section (p. 12, lines 13-17).

"The differences in air pollutant concentrations reported by the ACCMIP models reflect different treatments of atmospheric dynamics and chemistry, chemistry-climate interactions, and natural emissions in each model. Although there is likely a bias in estimating health effects using air pollutant concentrations from coarse resolution models (Li et al., 2015; Punger and West, 2013), particularly for PM2.5, we do not expect resolution to be an important factor for the differences in simulated concentrations across coarse resolution models."

*It would be nice to see a graph summarizing other paper's results and this one (with error bars) for projected mortality burdens and to discuss what could be learned from this comparison.*

As we now discuss more fully in the Introduction, there are several studies that are comparable, but there are many differences among these studies in the pollutants and health effects considered, the concentration-response functions used, the scenarios modeled and the time periods evaluated. Because of these large differences we suggest that a systematic comparison over all of the literature would be difficult and would not yield the meaning that the Referee asks for. We choose not to compare with the whole literature, but focus on comparing with Lelieveld et al. (2015), who used comparable health estimation methods but different future scenarios.